# Let Androids Dream of Electric Sheep: A Human-Inspired Image Implication Understanding and Reasoning Framework

## Abstract

Metaphorical comprehension in images remains a critical challenge for AI systems, as existing models struggle to grasp the nuanced cultural, emotional, and contextual implications embedded in visual content. While multimodal large language models (MLLMs) excel in basic Visual Question Answer (VQA) tasks, they exhibit a fundamental limitation on image implication tasks: contextual gaps that obscure the relationships between different visual elements and their abstract meanings. Inspired by the human cognitive process, we propose *Let Androids Dream (LAD)*, a novel framework for image implication understanding and reasoning. LAD addresses contextual missing through the three-stage framework: (1) **Perception**: converting visual information into rich and multi-level textual representations, (2) **Search**: iteratively searching and integrating cross-domain knowledge to resolve ambiguities, and (3) **Reasoning**: generating contextual-alignment image implication via explicit reasoning. Our framework with the lightweight GPT-4o-mini model achieves SOTA performance compared to 15+ MLLMs on English image implication benchmark and a huge improvement on Chinese benchmark, performing comparable with the GPT-4o model on Multiple-Choice Question (MCQ) and outperforms 36.7% on Open-Style Question (OSQ). Additionally, our work provides new insights into how AI can more effectively interpret image implications, advancing the field of vision-language reasoning and human-AI interaction. Our project is publicly available at https://anonymous.4open.science/r/Let-Androids-Dream-of-Electric-Sheep.

## 1 Introduction

> *Do androids dream of electronic sheep? The question actually has two levels: The first level is to ask if androids dream, and the second level is to ask if they dream of electronic sheep.*
>
> – Philip K. Dick (1968)

Metaphors are not just abstract concepts found in literature; they are also prevalent in our daily lives. For instance, when we say "time is money" or "life is a journey", we are using metaphors to convey complex ideas in a more contextual and understandable way. These metaphors highlight the integral role that metaphoric thinking plays in human communication. Just as we use metaphors to make sense of the world around us, we aim to enable AI to understand metaphors in a human-like manner. As established in "Metaphors We Live By" (Lakoff & Johnson, 2008), metaphors are not merely ornamental language devices but fundamental cognitive tools that allow us to conceptualize our surroundings. Metaphors possess characteristics such as systematicity, the creation of similarity, and imaginative rationality. Through cross-domain mapping, one concept can be used to comprehend another, allowing for a more insightful interpretation.

With the rapid advancement of large language models (LLMs), models such as OpenAI o1 (OpenAI, 2024b), DeepSeek-R1 (DeepSeek-AI, 2025), and QwQ (Team, 2024b) have demonstrated remarkable text-reasoning capabilities. However, a significant amount of knowledge in the real world cannot be fully represented by text alone. Visual information, for instance, contains a wealth of knowledge that is not easily captured through text. As a result, there has been a growing interest in integrating visual information into text-reasoning tasks. Compared to language, vision is inherently complex due to its diverse representation, subjective understanding, and difficulty in quantifying its data.

In recent years, vision-language reasoning models such as QVQ (Team, 2024a) and Grok-3-reasoning (xAI, 2025) have achieved outstanding performance. For example, Grok-3-reasoning model has reached a high score on math, code and vision-language reasoning benchmarks (Lightman et al., 2023; Lu et al., 2024; Wang et al., 2024; Yue et al., 2024). However, these models still struggle with image metaphor questions (Liu et al., 2024; Zhang et al., 2024). They tend to focus on the superficial elements of the image, neglecting the deeper connections and emotional expressions among them, as shown in Figure 1. It is important to note that these models excel at logical reasoning tasks, which are based on a different set of cognitive principles compared to image metaphor. Unlike VQA tasks that focus on concrete image comprehension, image metaphors require a stronger emphasis on abstract meaning and higher-order reasoning abilities. It is not a simple logical reasoning task and needs a different method to understand implications. It requires the model to grasp complex and abstract information, such as metaphors, symbols, and emotions in the image, rather than just concrete contents.

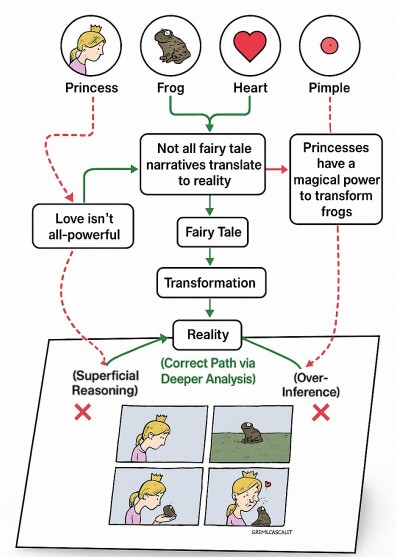

Figure 1: An image is worth a thousand words: For the image implication understanding task, different elements' combination lead to different thinking paths, but the correct path needs all elements with multiple reasoning thoughts.

Image implication tasks consist of two main aspects: understanding and generation. Understanding image implication is a more complex and challenging task than conventional images. It requires advanced cognitive abilities such as multi-hop reasoning and a sophisticated theory of mind (ToM), which are inherent to human cognition (Liu et al., 2024; Zhang et al., 2024). Compared to understanding, generating implication is even more difficult. The challenge stems from the lack of contextual understanding of the key elements and internal relationships of the image. This lack of context hinders our ability to decipher the intended message or to create images that effectively convey specific meanings. Without the background of cultural, or environmental context, the significance of key visual components remains elusive, impeding both interpretation and creative expression.

Existing methods for image metaphor understanding mainly fall into two categories: explicit mapping and implicit reasoning. Explicit mapping creates a link between metaphor ontology and visual representation. For example, the CLOT method (Zhong et al., 2024) uses this mapping to understand image metaphors. However, it struggles with complex many-to-many mappings and dynamically changing cultural backgrounds. Implicit reasoning relies on the model's ability to reason without explicit mapping. For instance, the C4MMD method (Xu et al., 2024b) uses an untrained reasoning approach. Despite its potential, it still faces challenges in handling complex metaphor understanding tasks, especially in situations involving multimodal information and cultural backgrounds.

To address these problems, inspired by how humans (possibly) understand metaphors, we find that the essence of the difficulty in metaphor understanding and generation is contextual missing. Therefore, we propose a novel framework that more closely aligns with human cognitive processes for metaphor interpretation. Our framework first transforms visual information into textual representations and then iteratively searches to enrich these representations with out-of-domain knowledge, enabling deeper inferential reasoning. Experiments from both Multiple-Choice Question and Open-Style Question consistently verify the superiority of the proposed framework.

*Our key contributions are listed as follows:*

- We systematically analyze image implication tasks and find the difficulty of the metaphor understanding and reasoning task lies in contextual missing. From the perspective of human cognition, we proposed a new direction for solving these tasks – Contextual Alignment.

- We propose a novel human-inspired three-stage framework Let Androids Dream (LAD) for image implication understanding and reasoning, including Perception, Search and Reasoning. Our LAD implements the lightweight GPT-4o-mini model to achieve SOTA on English image implication

benchmark and a huge improvement on Chinese image implication benchmark, comparable with the GPT-4o model and other top closed-source models on Multiple-Choice Question (MCQ).

- We design the challenging Open-Style Question (OSQ) with comprehensive metric to automatic evaluate the image implication tasks. This metric aligns 95.7% with human annotations, making it more suitable for diverse evaluation. Our LAD outperforms the GPT-4o model 36.7% on OSQ.

## 2 RELATED WORK

### 2.1 IMAGE IMPLICATION

Image implication encompasses various cognitive aspects, including humor, sarcasm, and broader metaphorical understanding. Early research focused on specialized aspects, such as humor recognition (Hessel et al., 2023; Horvitz et al., 2024) and sarcasm detection (Desai et al., 2022). As the rapid development of large language models brings new opportunities for analyzing image implication, we need more comprehensive evaluation frameworks. DeepEval (Yang et al., 2024b) provided a systematic taxonomy of image implications. Subsequently, II-Bench (Liu et al., 2024) emerged as the first English image implication benchmark, followed by CII-Bench (Zhang et al., 2024), which extended this benchmark to Chinese images. Implication understanding requires sophisticated multi-hop reasoning and theory of mind (ToM) capabilities (Liu et al., 2024; Zhang et al., 2024). Existing approaches fall into two categories: explicit mapping and implicit reasoning. The first approach, represented by CLOT (Zhong et al., 2024), constructs mappings between metaphor ontologies and visual representations. However, this approach faces key challenges: metaphorical relationships have complex many-to-many mappings that are difficult to formalize, and cultural references are too dynamic for static mappings. The second approach, exemplified by C4MMD (Xu et al., 2024b), employs training-free CoT reasoning. Despite its promise, this approach struggles with the complex nature of metaphorical understanding, which surpasses traditional reasoning. The large search space for out-of-domain reasoning and changing cultural contexts limits its effectiveness. To address this, we propose a novel methodology that transforms visual information into texts and iteratively enriches them with out-of-domain knowledge, better aligning with human cognitive processes.

### 2.2 VISION-LANGUAGE REASONING

The rapid advancement of LLMs has demonstrated remarkable text reasoning capabilities, as evidenced by models such as o1 (OpenAI, 2024b), DeepSeek-R1 (DeepSeek-AI, 2025), and QwQ (Team, 2024b; Yang et al., 2024a). However, real-world knowledge often transcends textual representation, with visual information encapsulating substantial knowledge that pure language models cannot access. For example, images inherently contain rich, multi-layered information that often resists straightforward description, including spatial relationships, contextual nuances, and implicit knowledge that humans process intuitively. This limitation has driven research toward integrating them into text-based reasoning frameworks. Current research has developed three primary approaches to incorporate visual information into model reasoning: 1) Comprehensive MLLM Description: This approach treats visual content as a text grounding problem, as demonstrated by LLAVA-CoT (Xu et al., 2024a) and Mulberry (Yao et al., 2024). 2) Multi-turn MLLM Interaction: Models like VoCoT (Li et al., 2024b) and V* (Wu & Xie, 2023) employ iterative question-answering to extract fine-grained visual information at various levels of detail. 3) Tool-augmented Reasoning: Frameworks such as Visual Sketchpad (Hu et al., 2024) and Whiteboard-of-Thought (Menon et al., 2024) leverage tool-based approaches to modify images and augment reasoning with prior knowledge embedded in tools. However, the challenge for image implication understanding task is typically not a deficit in the image's content but a "contextual missing"—the lack of external cultural, social, or historical knowledge for interpretation. Therefore, these methods actively alter the visual input (e.g., by sketching or editing), which are not suitable for implication understanding.

## 3 METHOD

Inspired by the human cognitive process, we introduce a new paradigm for solving image implication tasks – Contextual Alignment. We have a detailed discussion for this point in Section 1 and Section 5. Therefore, we propose Let Androids Dream (LAD), a novel framework for image implication understanding and reasoning. This framework operates through the three-stage framework, as shown in Figure 2: (1) **Perception**: converting visual information into rich and multi-level texts, (2) **Search**: iteratively searching and integrating cross-domain knowledge to resolve ambiguities, and (3) **Reasoning**: generating contextual-alignment analysis via explicit reasoning.

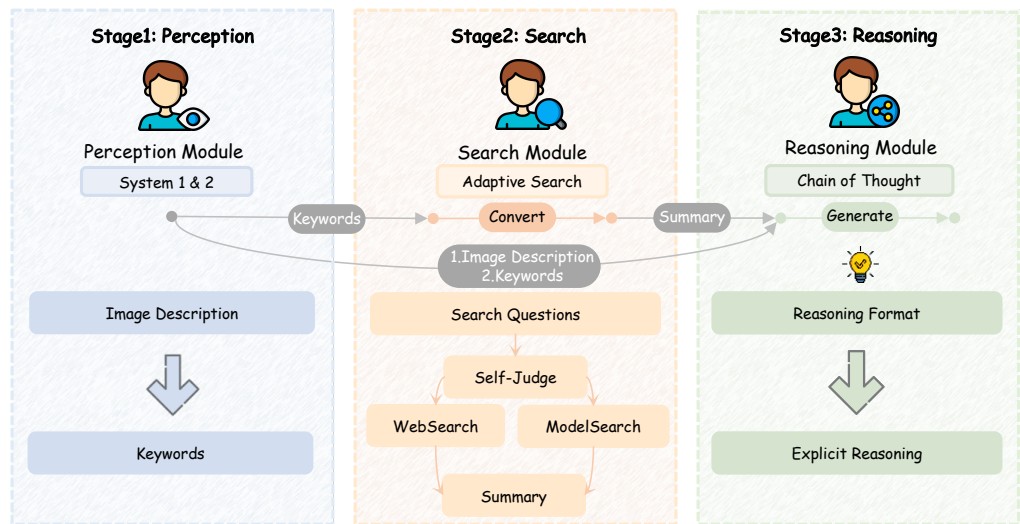

Figure 2: The general framework of Let Androids Dream (LAD), which includes three stages: (1) Perception: converting raw visual information into rich and multi-level textual representations, (2) Search: iteratively searching and integrating cross-domain knowledge to resolve ambiguities, and (3) Reasoning: generating context-alignment image implication interpretations via explicit reasoning.

## 3.1 STAGE I: PERCEPTION

The initial stage, *Perception*, aims to transform raw visual inputs into structured, hierarchical textual representations, mirroring the human cognitive process of initial intuition-driven observation and subsequent identification of key elements. This stage operates in a manner analogous to human System 1 (intuitive, holistic processing) and System 2 (analytical, focused processing).

First, we utilize MLLM to process the input image and produce a detailed textual narrative. This description captures coarse-grained visual information, including discernible text within the image, prominent colors, overall layout, and salient objects or entities. This step provides a foundational understanding of image contents. Following this, we derive a fine-grained keyword set. The MLLM condenses the above description into a concise set of approximately 7 keywords. These keywords are specifically chosen to encapsulate critical aspects relevant to implication understanding, such as the perceived emotion, the domain or context (e.g., social, cultural) and any rhetorical devices that might be visually suggested. Keywords also re-emphasize crucial textual elements or entities identified in the description. This two-tiered representation, comprising a rich description and focused keywords, provides a robust foundation for the subsequent *Search* and *Reasoning* stages. The keywords, in particular, serve as vital cues for guiding the knowledge retrieval in stage II.

## 3.2 STAGE II: SEARCH

The *Search* stage addresses semantic ambiguities and enhances contextual comprehension by iteratively retrieving and integrating critical cross-domain knowledge for interpreting image implications. It employs adaptive search, which dynamically selects the most appropriate search method. The process is systematically organized into three main phases: Plan, Search, and Summary.

**1. Plan**: The process begins by formulating targeted search queries. Using the keywords generated in Stage I, the MLLM, guided by a prompt specifically designed for image implication tasks, generates five different levels of search questions. These questions aim to uncover latent meanings, cultural references, or background information pertinent to the image implications.

**2. Search**: This phase executes the search based on the generated questions, employing the Self-Judge mechanism to determine the optimal search strategy for each question.

(a) **Self-Judge**: Our system uses the MLLM as a router, scoring queries based on criteria like knowledge popularity and specificity. Queries with high scores, indicating a need for current or niche information, are routed to WebSearch, while those answerable from general knowledge use ModelSearch. This ensures efficient and comprehensive knowledge coverage.

(b) **ModelSearch**: For questions suitable for internal retrieval, ModelSearch uses the MLLM's parametric memory. With a specialized prompt, the model generates answers from its pre-trained knowledge, efficiently recalling established facts or common concepts.

(c) **WebSearch**: For questions requiring external, dynamic, or highly specific information, Web-Search is invoked. Inspired by LLM search methods like MindSearch (Chen et al., 2024), but focusing on image implication tasks, our WebSearch component first employs the planner. The planner, acting as a high-level strategist, decomposes the initial search question into a series of more granular sub-questions. These sub-questions are structured into a directed acyclic graph (DAG), simulating a multi-step, exploratory information-seeking process. Subsequently, the searcher executes this plan. It performs hierarchical information retrieval for each sub-question from the internet, gathering relevant snippets and facts. This multi-agent method, with distinct planner and searcher modules, allows for parallel processing and dynamic refinement of the search strategy. The retrieved information for sub-questions is then synthesized to answer the original search question. This ensures access to recent developments and a broad spectrum of public knowledge, crucial for understanding contemporary image implications.

**3. Summary**: The raw outputs from the Search phase are refined into a concise search summary.

(a) **RankSummary:** The set of five question-answer pairs is evaluated. The MLLM ranks these pairs based on their relevance to understanding the core implication of the original image. The top three most relevant question-answer pairs are selected.

(b) **RefineSummary:** The selected pairs are further processed. The MLLM, guided by the ranking reason from the ranking step, rewrites and consolidates these pairs. This involves removing irrelevant or redundant information, reconciling diverse pieces of information, and potentially supplementing details to create a single, optimized, and concise search summary. This final summary serves as the enriched contextual input for Stage III.

### 3.3 STAGE III: REASONING

The final stage, *Reasoning*, performs explicit reasoning to derive contextually grounded interpretations of image implications. This stage synthesizes all previously gathered information — the hierarchical textual representations from Stage I (descriptions and keywords) and the domain-enriched knowledge from Stage II — into a coherent implication framework. For image implication tasks, we employ a specific reasoning format. The MLLM is prompted to articulate its reasoning trajectory using designated markers, such as "<think> ...</think>" special tokens. Within these markers, the model explicitly lays out its step-by-step reasoning process, connecting the visual cues, keywords, and external knowledge to arrive at the final image implication analysis and explanation. This domain-specific CoT method not only guides the model towards a more grounded output, but also makes the inferential pathway transparent. The framework ultimately generates a contextually-aligned implication understanding that emerges from the integration of semantic inputs and cross-domain knowledge, formalizing the LAD system's capacity for evidence-based visual reasoning.

### 3.4 LAD PIPELINE

Our LAD framework functions as a sequential pipeline, integrating three stages shown in Figure 2 and Algorithm 1. **Stage I (Perception)** starts the process. It takes an input image, uses the MLLM to generate a detailed image description, and extracts seven key keywords. The outputs are the image description and the keywords. **Stage II (Search)** uses these keywords as input. The MLLM converts them into five search questions. A self-judge mechanism directs these questions to ModelSearch or WebSearch. The top three related question-answer pairs are selected and refined into a concise search summary. **Stage III (Reasoning)** receives the original image, the description and keywords from Stage I, and the search summary from Stage II. The MLLM integrates these inputs and generates the final image implication through a structured reasoning process. This implication represents the culmination of the LAD pipeline's understanding and reasoning about the input image.

## 4 EXPERIMENT

### 4.1 BASELINES

**Models.** To comprehensively compare with LAD, we carefully select a diverse range of MLLMs, with the aim of covering a wide spectrum of model characteristics and scales. These models span parameter sizes from 7B to 300B, ensuring that models of varying complexity and capability are

| Model | Multiple-Choice Question | | Open-Style Question | |
|---|---|---|---|---|
| | en | zh | en | zh |
| *General Models* | | | | |
| Qwen2.5-VL-7B (Bai et al., 2025) | 46% | 40% | 2.34 | 2.58 |
| DeepSeek-VL2 (Wu et al., 2024) | 46% | 36% | 2.82 | 2.86 |
| GLM-4.1V-8B (Zhipu.ai, 2024) | 60% | 52% | 2.60 | 2.96 |
| Gemini-2.0-flash (Team, 2023) | 70% | 68% | 1.60 | 3.12 |
| Qwen2.5-VL-72B (Bai et al., 2025) | 72% | 56% | 1.56 | 3.12 |
| InternVL3-78B (Zhu et al., 2025) | 70% | **74%** | 3.42 | 3.70 |
| GLM-4V-plus (Zhipu.ai, 2024) | 64% | 64% | 3.01 | 3.12 |
| Gemini-2.0-pro (Team, 2023) | 68% | 62% | 1.66 | 3.18 |
| Grok-3 (xAI, 2025) | 66% | 64% | 3.24 | 2.96 |
| Claude-3.5-Sonnet (Anthropic, 2024) | 68% | 62% | 3.22 | 3.78 |
| GPT-4o (OpenAI, 2024a) | **74%** | 58% | 2.94 | 3.76 |
| GPT-4.1 (OpenAI, 2024a) | **74%** | 62% | 3.30 | **3.92** |
| *Vision-language Reasoning Models* | | | | |
| Gemini-2.0-flash-thinking (Team, 2023) | 64% | 68% | 1.66 | 2.84 |
| QVQ-72B (Team, 2024a) | 62% | 56% | 3.10 | 3.42 |
| Doubao-1.5-thinking-vision-pro (Seed, 2025) | 66% | 66% | 3.16 | 3.90 |
| Grok-3-reasoning (xAI, 2025) | **74%** | 64% | 3.06 | 2.92 |
| *Our Method* | | | | |
| GPT-4o-mini (OpenAI, 2024a) | 44% | 42% | 2.98 | 3.36 |
| + LAD (Stage I + III) | 68% ↑ | 44% ↑ | 3.84 ↑ | 3.58 ↑ |
| + LAD (Stage I + II + III) | **74%** ↑ | 52% ↑ | **4.02** ↑ | 3.66 ↑ |
| Improv. | +30 (68.2%) | +10 (23.8%) | +1.04 (34.9%) | +0.3 (8.9%) |

Table 1: Overall results of different models on Multiple-Choice Question and Open-Style Question. The best-performing model in each category is **in-bold**, and the second best is underlined.

thoroughly assessed. In selecting the models, we focus on the following key aspects: 1) General and Reasoning models, 2) Open-Source and Closed-Source models, and 3) model parameter scaling law. The settings is in Appendix B, and the full prompt is in Appendix H.

**Evaluation.** Our evaluation utilizes two comprehensive image implication benchmarks, II-Bench (Liu et al., 2024) and CII-Bench (Zhang et al., 2024), both featuring Multiple-Choice Question (MCQ). Furthermore, we manually construct the high-level benchmark by selecting 100 high-quality, diverse and representative images from varied image types like illustrations and comics. The detail statistic is in Appendix D. And we measure accuracy by comparing the model's selected option to the ground truth. Aware of potential MCQ biases (Li et al., 2024a; Zheng et al., 2024) and the greater difficulty of generation over judgment tasks, we design the challenging Open-Style Question (OSQ). It uses the same images with the fixed question: "What is the implication in this image?". And we use GPT-4o with a specialized evaluation metric as evaluators, validated by multiple human consistency checks. The representative main experiments with the high-level benchmark are shown as follows, and the large-scale generalization experiments with the full II-Bench (1399 English) and CII-Bench (800 Chinese) are detailed in Appendix E, wherein LAD consistently shows significant performance. We also conduct a further analysis about experiments' findings in Appendix F.

## 4.2 MULTIPLE-CHOICE QUESTION

### 4.2.1 IMPLEMENTATION DETAILS

Our benchmark includes diverse images such as comics, posters, Internet memes, and Chinese traditional artworks, all rich in visual information and cultural significance. Each image is paired with one question, each offering six options with only one correct answer. The question is "What is the implication in this image?" (mostly) or different levels of image understanding, such as overarching interpretation and nuanced details. A case study of different methods on MCQ is in Figure 3.

### 4.2.2 RESULTS AND ANALYSIS

Table 1 shows MCQ results for various MLLMs on our high-level benchmark. The LAD framework is highly effective, achieving SOTA performance with the lightweight GPT-4o-mini model. In English MCQ, it matches closed-source models like GPT-4o, GPT-4.1, and Grok-3-reasoning (74%), and significantly outperforms Claude-3.5-Sonnet and Gemini-2.0-pro by 9%. For Chinese MCQ, it achieves comparable results to GPT-4o, while substantially surpassing DeepSeek-VL2 by 44.4%. Compared to the base GPT-4o-mini model, our framework shows major improvements: 68.2% in

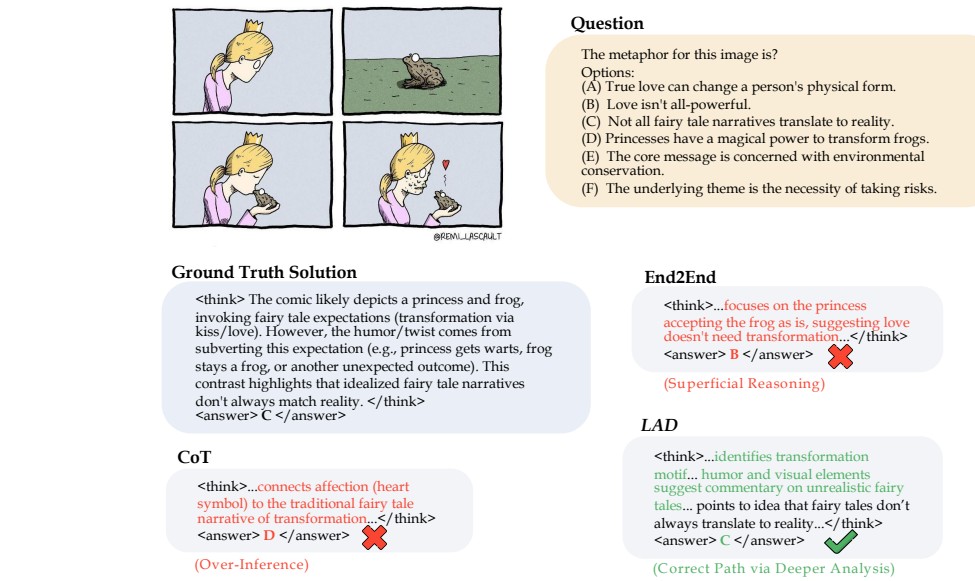

Figure 3: A case study of different methods on Multiple-Choice Question. The *End2End* method shows superficial reasoning and the *CoT* method shows over-inference, while our *LAD* framework shows the correct path via more contextual alignment analysis.

English and 23.8% in Chinese, far beyond other open-source and reasoning models. Notably, we observe that reasoning models offer little advantage over general models on image implication task, with similar accuracy across categories. This suggests that current RL-based reasoning methods have limited generalization for image implication understanding, highlighting its distinct complexity compared to basic VQA tasks and classic logical reasoning domains like math and code.

## 4.3 OPEN-STYLE QUESTION

### 4.3.1 IMPLEMENTATION DETAILS

**Evaluation Metric.** To comprehensively assess MLLMs' understanding of image implication, we develop a multifaceted evaluation metric. This metric is designed to probe both the surface-level information readily apparent in the image and the deeper emotion, domain and rhetorical skills that inform its creation and interpretation. Our evaluation metric encompasses five key perspectives: *Surface-level Information*, *Emotional Expression*, *Domain and Context*, *Rhetorical Skills*, and *Deep Implications*. For each perspective, we give its detailed description in Figure 4 in Appendix H.

**MLLM-based Automatic Evaluation.** To evaluate image implication understanding in MLLMs, we develop an MLLM-based evaluation standard based on evaluation metrics, as illustrated in Figure 4 in Appendix H. Our experiment utilizes the high-level benchmark, with human-written descriptions and implications as ground truth. We choose the same MLLMs with MCQ experiment to generate image implications for these images, which are subsequently scored using GPT-4o and our evaluation standard. The evaluation prompt is in Appendix H. To validate the model's scoring efficacy, we enlist 16 PhD students and researchers well-versed in English and Chinese metaphorical imagery to independently score the dataset. The human-model scoring consistency reached **95.7%**, affirming the method's validity. The detailed human-model consistency study is in Appendix C.

### 4.3.2 RESULTS AND ANALYSIS

Table 1 shows OSQ results for various MLLMs on our high-level benchmark. The LAD framework demonstrates exceptional effectiveness, achieving SOTA performance with the lightweight GPT-4o-mini model. In English OSQ, our framework substantially outperforms closed-sourced models like GPT-4o by 36.7% and Claude-3.5-Sonnet by 24.8%. For Chinese OSQ, while slightly below top closed-sourced models like GPT-4.1 and Doubao-1.5-thinking-vision-pro, our method still significantly surpasses Gemini-2.0-pro by 15.1% and DeepSeek-VL2 by 30%. The enhancement over the GPT-4o-mini is particularly noteworthy, with improvements of 34.9% for English and 8.9% for

| Model | Multiple-Choice Question | | Open-Style Question | |
|---|---|---|---|---|
| | en | zh | en | zh |
| *GPT-4o-mini* | | | | |
| w/o CoT | 44% | 42% | 2.98 | 3.36 |
| Standard CoT | 50% ↑ | 42% | 3.10 ↑ | 3.28 ↓ |
| LAD-CoT | **68%** ↑ | **44%** ↑ | **3.84** ↑ | **3.58** ↑ |

Table 2: Results of different CoT methods.

| Model | MCQ | | OSQ | | Model | MCQ | | OSQ | |
|---|---|---|---|---|---|---|---|---|---|
| | en | zh | en | zh | | en | zh | en | zh |
| *Grok-3* | | | | | *Qwen2.5-VL-7B* | | | | |
| w/o search | 66% | 64% | 3.24 | 2.96 | w/o LAD | 46% | 40% | 2.34 | 2.58 |
| Grok-Search | **72%** ↑ | 64% | 3.25 ↑ | 2.92 ↓ | w/ LAD | **64%** ↑ | **46%** ↑ | **3.64** ↑ | **3.36** ↑ |
| *GPT-4o* | | | | | *Qwen2.5-VL-72B* | | | | |
| w/o search | 74% | 58% | 2.94 | 3.76 | w/o LAD | 72% | 56% | 1.56 | 3.12 |
| Perplexity (pro) | **80%** ↑ | **66%** ↑ | 2.88 ↓ | 3.28 ↓ | w/ LAD | **76%** ↑ | **62%** ↑ | **3.62** ↑ | **3.68** ↑ |
| *GPT-4o-mini* | | | | | *GPT-4o* | | | | |
| w/o search | 68% | 44% | 3.84 | 3.58 | w/o LAD | 74% | 58% | 2.94 | 3.76 |
| GPT-Search | 72% ↑ | 48% ↑ | 3.62 ↓ | 3.34 ↓ | w/ LAD | **80%** ↑ | **66%** ↑ | **4.14** ↑ | **4.26** ↑ |
| LAD-Search | **74%** ↑ | **52%** ↑ | **4.02** ↑ | **3.66** ↑ | | | | | |

Table 4: Results of different base models.

Table 3: Results of different search methods.

Chinese, far exceeding other open-source and reasoning models. Unlike MCQ results, we observe significant performance disparities between reasoning and general models on OSQ, highlighting the distinct challenges of image implication generation. Interestingly, several models (e.g., Qwen2.5-VL-72B) exhibit substantial performance gaps between MCQ and OSQ. Upon manual examination of model outputs, we attribute this to potential overfitting to multiple-choice formats and insufficient exposure to open-style generation tasks. In addition, LLMs or even MLLMs may not genuinely understand the questions but rather predict options as answers, introducing evaluation bias and demonstrating sensitivity to option positioning (Zheng et al., 2024).

## 4.4 ABLATION STUDY

### 4.4.1 STAGE I (PERCEPTION) AND STAGE III (REASONING)

We incorporate LAD's Stage I (Perception) and Stage III (Reasoning), collectively LAD-CoT. It shows significant improvements in Table 1, with GPT-4o-mini scores increasing from 44% to 68% (English) on MCQ, and from 2.98 to 3.84 (English) and 3.36 to 3.58 (Chinese) on OSQ. As shown in Table 2, standard CoT yields minor gains on English tasks (MCQ: 44% to 50%; OSQ: 2.98 to3.10) but shows no improvement or a slight decline on Chinese tasks (MCQ: 42% to 42%; OSQ: 3.36 to 3.28). In contrast, LAD-CoT substantially outperforms both the baseline and standard CoT across all types. For instance, LAD-CoT achieves 68% on English MCQ while standard CoT only 50%, and a score of 3.84 on English OSQ compared to 3.10 for standard CoT. These findings highlight the superior efficacy of our LAD-CoT for image implication over standard CoT methods. A case study of various CoT on MCQ is in Figure 3. The standard CoT prompt and other details is in Appendix H.

### 4.4.2 STAGE II (SEARCH)

We analyze LAD's Stage II (Search), termed LAD-Search. It shows significant improvements in Table 1, with GPT-4o-mini's MCQ scores rising from 68% to 74% (English) and 44% to 52% (Chinese), and OSQ scores from 3.84 to 4.02 (English) and 3.58 to 3.66 (Chinese). Compared with Grok-3-search, GPT-4o-mini-search-preview, and GPT-4o with Perplexity.ai (Pro), results are shown in Table 3. GPT-Search boosts GPT-4o-mini's MCQ scores yet drops OSQ performance (English OSQ: 3.84 to 3.62, Chinese OSQ: 3.58 to 3.34). Grok-Search on Grok-3 offers limited gains, mainly in English MCQ, with inconsistent Chinese results and minimal OSQ improvement. Perplexity.ai search with GPT-4o greatly raises MCQ accuracy but sharply lowers OSQ scores (English OSQ: 2.94 to 2.88, Chinese OSQ: 3.76 to 3.28). In contrast, LAD-Search consistently improves both MCQ and the more challenging OSQ. This highlights its superior ability to integrate external

knowledge for implication understanding, outperforming other search methods in open-style reasoning. General search methods may introduce too much and too diverse information, which is not useful for complex and subjective problems like image metaphors. Our LAD-Search effectively addresses this issue through algorithm design, thus achieving better results.

### 4.4.3 DIFFERENT BASE MODELS

To demonstrate the generalizability of our LAD framework beyond the GPT-4o-mini model, we conduct experiments on other base models, including the open-source Qwen2.5-VL series and the closed-source GPT-4o. As the Table 4 shows, applying LAD framework significantly improves all models' performance on both MCQ and OSQ tasks, confirming that our framework is not model-specific and provides a robust and generalizable way to enhancing image implication understanding.

## 5 DISCUSSION

### 5.1 HUMAN COGNITIVE THEORY OF LET ANDROIDS DREAM

LAD is analogous to human cognitive strategies, not a direct neuroscientific replica. We aim to create a system that reasons transparently and aligns with human problem-solving methods, not to perfectly simulate the human brain. Our framework is inspired by established cognitive theories: (1) Dual-Process Theory (Evans, 2003): The Perception stage reflects the interplay between System 1 (fast, intuitive, holistic impression) and System 2 (slow, analytical identification of key elements), and (2) Active Information-Seeking Theory (Ikoja-Odongo & Mostert, 2006; Wilson, 2009): The Search stage is like humans actively seeking external infomation to resolve ambiguities. When we encounter unfamiliar memes or cultural references, we often "Google it" to supplement our knowledge. Our WebSearch module simulates this deliberate information-foraging behavior.

### 5.2 HOW TO LET ANDROIDS DREAM? PERCEPTION AND REASONING

The question "How to Let Androids Dream?" metaphorically addresses the core challenge of enabling AI systems to interpret the nuanced implications in images. Our framework tackles this by first emulating human-like perception (Stage I), converting raw visual input into rich, multi-level textual representations, including detailed descriptions and key keywords. These keywords capture not only objects and scenes but also potential emotional tones, relevant domains (e.g., cultural, social, political), and discernible rhetorical devices. Subsequently, LAD's Stage III employs a structured CoT process. This reasoning guides the model to systematically connect perceived visual elements with retrieved contextual knowledge, building a coherent understanding. This is crucial because, as our experiments (Section 4) and recent work on social reasoning (Kim et al., 2025) show, comprehending implications goes beyond basic VQA tasks and classic logic; it involves sophisticated social reasoning and interpreting contextual cues often missed by MLLMs.

### 5.3 HOW TO DREAM OF ELECTRIC SHEEP? SEARCH

Expanding on analysis skills, "How to Dream of Electric Sheep?" explores how AI can accurate and specific image implications—the metaphorical 'electric sheep'. LAD's Stage II (Search) is crucial. This stage notes that visual meanings, especially in metaphors, often depend on external info like culture, history, or current events, which MLLMs' static knowledge may lack. LAD's search mechanism—forming queries from keywords and choosing between ModelSearch and WebSearch via Self-Judge—enriches initial perceptions with cross-domain knowledge. This iterative process, particularly for popular metaphors or vague visuals, widens the model's interpretive scope. By adding essential context, Search helps LAD move past surface-level interpretations to accurately grasp subtle image meanings, as shown in its strong Open-Style Question (OSQ) performance.

## 6 CONCLUSION

Understanding image implications remains challenging for MLLMs, mainly due to contextual missing. Our work introduces Let Androids Dream (LAD), a novel three-stage framework: Perception, Search, and Reasoning. Inspired by human cognitive processes, this framework is designed to achieve contextual alignment by explicitly integrating visual interpretation with external knowledge retrieval. We conduct comprehensive experiments to demonstrate its effectiveness. Utilizing the lightweight GPT-4o-mini, LAD achieves top results on implication benchmarks, performing comparable or even surpassing GPT-4o and other top closed-source models, particularly on challenging OSQ. In summary, LAD bridges the gap between superficial perception and reasoning in multimodal AI systems, offering a promising direction for contextual-alignment vision-language reasoning.

## REPRODUCIBILITY STATEMENT

To ensure that our work can be effectively reproduced by other researchers, we have dedicated significant effort to enhancing reproducibility. In order to facilitate the full reproducibility of our results, we have taken the following steps: 1) included comprehensive technical details in Section 4 and Appendix A, B, D, H, and 2) published both our code and data with thorough documentation at https://anonymous.4open.science/r/Let-Androids-Dream-of-Electric-Sheep. We are committed to maintaining our GitHub repository, engaging in discussions with other researchers, and contributing to the broader VLM community.

## ETHICS STATEMENT

The LAD framework aims to enhance AI's nuanced understanding of image implications, a crucial aspect of human-like cognition. We acknowledge that advanced interpretative capabilities carry ethical considerations, including potential biases inherited from underlying MLLMs or training data, and the risk of misuse in generating or interpreting content. Our use of public benchmarks promotes transparency in evaluation. We are committed to fostering responsible development and encourage continued research into robust safeguards and ethical AI practices within multimodal reasoning to ensure beneficial applications.

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

## A  ALGORITHM

---

**Algorithm 1** Let Androids Dream (LAD)

---

**Input:** Image $IMG$, Task $T_{MCQ}$, Task $T_{OSQ}$
**Output:** Answer $A_{MCQ}$, Answer $A_{OSQ}$

  // Stage I: Perception
1  $img\_dep \leftarrow$ MLLM.Perception($IMG$)                              /* Gen. description. */
2  $keywords \leftarrow$ MLLM.Perception($img\_dep$)                           /* Gen. 7 keywords */
  // Stage II: Search
3  $search\_qs \leftarrow$ MLLM.Plan($keywords$)           /* 5 questions for image implication */
4  $all\_qa \leftarrow \emptyset$
5  **for** *each q in search_qs* **do**
6     $strategy \leftarrow$ MLLM.Self-Judge($q$)
7     **if** $strategy =$ **then**
8        $answer \leftarrow$ WebSearch($q$)                      /* External knowledge */
9     **end**
10    **else if** $strategy =$ **then**
11       $answer \leftarrow$ ModelSearch($q$)                 /* Parametric knowledge */
12    **end**
13    $all\_qa$.add(($q, answer$))
14  **end**
15  $search\_sum \leftarrow$ MLLM.Summary($img\_dep, all\_qa$)         /* Rank top-3, refine */
  // Stage III: Reasoning
16  $A_{MCQ} \leftarrow$ MLLM.Reasoning($IMG, img\_dep, keywords, search\_sum, T_{MCQ}$)    /* Explicit CoT */
17  $A_{OSQ} \leftarrow$ MLLM.Reasoning($IMG, img\_dep, keywords, search\_sum, T_{OSQ}$)    /* Explicit CoT */
18  **return** $A_{MCQ}, A_{OSQ}$

---

19  **Function** *WebSearch*($q$)
    // Planner: Decompose query
20    $sub\_qs \leftarrow$ MLLM.RewriteQuery($q$)
    // Searcher: Hierarchical retrieval
21    $snippets \leftarrow$ .BatchQuery($sub\_qs$)                /* Titles, summaries, URLs */
22    $sel\_urls \leftarrow$ MLLM.SelectPages($snippets, q$)
23    $content \leftarrow$ .FetchContent($sel\_urls$)
    // Summarizer: Generate answer
24    $summary \leftarrow$ MLLM.Summary($content, q$)
25    **return** *summary*

---

## B  EXPERIMENT SETUP

We use the lightweight GPT-4o-mini-0718 (OpenAI, 2024a) with LAD framework in experiments. We set the model temperature as 0.5 and top_p as 0.9 in MCQ experiments, and temperature as 0.7 and top_p as 0.9 in OSQ experiments. Additionally, we set the evaluation model GPT-4o temperature as 0 and evaluate more than three times to get the average score in OSQ experiments. All experiments are conducted on NVIDIA A800 GPUs.

## C  HUMAN-MODEL CONSISTENCY STUDY

To validate our automated OSQ evaluation based on the GPT-4o model, we conduct a human-model consistency study. We construct a dedicated dataset by randomly selecting 25 images with questions each from our English and Chinese OSQ. We recruit 16 PhD students and researchers, all proficient in both English and Chinese and experienced with metaphorical imagery, to independently score the model responses. Their evaluations are based on ground truth answers and the detailed scoring standard. We calculate human inter-annotator agreement by averaging the scores for each response after discarding the highest and lowest individual scores. This process yields the consistency of 94.8% for Chinese and 96.5% for English. The average human-model scoring consistency reached 95.7%, affirming the method's validity for assessing image implication comprehension.

## D  STATISTICS

We manually construct the high-level benchmark by selecting 100 high-quality, diverse and representative images from II-Bench (Liu et al., 2024) and CII-Bench (Zhang et al., 2024). The general statistic is in Table 5.

| Statistics of English Images | | Statistics of Chinese Images | |
|---|---|---|---|
| Society | 21 (42%) | Life | 13 (26%) |
| Life | 16 (32%) | Art | 13 (26%) |
| Art | 6 (2%) | Society | 12 (24%) |
| Psychology | 4 (8%) | Chinese Traditional Culture | 6 (12%) |
| Others | 3 (6%) | Environment | 5 (10%) |
| | | Politics | 1 (2%) |
| Multi-panel Comic | 16 (32%) | | |
| Single-panel Comic | 9 (18%) | Illustration | 15 (30%) |
| Illustration | 5 (10%) | Single-panel Comic | 10 (20%) |
| Meme | 5 (10%) | Poster | 8 (16%) |
| Poster | 5 (10%) | Meme | 8 (16%) |
| Painting | 5 (10%) | Painting | 6 (12%) |
| Logo | 5 (10%) | Multi-panel Comic | 3 (6%) |

Table 5: General statistics of the high-level benchmark.

## E  GENERALIZATION EXPERIMENT

| Model | Multiple-Choice Question | | Open-Style Question | |
|---|---|---|---|---|
| | II-Bench (1399) | CII-Bench (800) | II-Bench (1399) | CII-Bench (800) |
| GLM-4.1V-8B | 70.0% | 46.3% | 2.83 | 3.06 |
| GPT-4o-mini | 63.5% | 35.6% | 2.93 | 3.29 |
| InternVL3-78B | 78.2% | **64.0%** | 3.68 | 4.06 |
| GPT-4o | 72.6% | 54.1% | 3.86 | 4.06 |
| Claude-3.5-Sonnet | 80.9% | 54.1% | 3.51 | 3.84 |
| LAD (GPT-4o-mini) | **81.2%** ↑ | 53.8% ↑ | **4.22** ↑ | **4.31** ↑ |

Table 6: Results of different models on full benchmarks. The best-performing model in each category is **in-bold**.

We conduct the large-scale experiments with the representative and top-performing models, including Closed-Source models GPT-4o and Claude-3.5-Sonnet, as well as the Open-Source model GLM-4.1V-8B, on the full benchmarks: II-Bench (1,399 English) and CII-Bench (800 Chinese) for both MCQ and OSQ tasks.

As the results in Table 6 show, our LAD framework's significant performance gains are consistent on these much larger datasets. Notably, by applying LAD, the lightweight GPT-4o-mini significantly surpasses the much larger GPT-4o and Claude-3.5-Sonnet. Compared with the baseline GPT-4o-mini model, we can find that: (1) On the large-scale English benchmark (II-Bench), our LAD framework improves the GPT-4o-mini score from 63.5% to 81.2% on MCQ and 2.93 to 4.22 on OSQ. This is a substantial absolute increase of 17.7% (27.9% relative improvement) and 1.29 (44% relative improvement). (2) The gains on the large-scale Chinese benchmark (CII-Bench) are even more pronounced. LAD boosts performance from 35.6% to 53.8% on MCQ and 3.29 to 4.31 on OSQ, representing an absolute increase of 18.2% (51.1% relative improvement) and 1.02 (31% relative improvement).

This robust improvement is consistent with the trend we observed and reported on our high-level benchmark (smaller 100-image dataset, 50 English and 50 Chinese) in Table 1. While the exact percentages differ due to the varying scales and baselines of the datasets, the key takeaway is that the significant positive impact of the LAD framework is undeniable across both small and large-scale evaluations. This analysis confirms that our framework's benefits are not an artifact of a small

test set but are indeed robust and generalizable. It also reflects the reliability and high quality of our manually curated high-level benchmark.

## F  Further Analysis on Method and Experiments

### F.1  Analysis of Let Androids Dream Success

Our analysis points to two primary failure modes for baseline models, which Let Androids Dream (LAD) is designed to mitigate. These are illustrated in Figure 1 and the case study in Figure 3:

**1. Superficial Reasoning**: This occurs when a model only processes the literal, surface-level elements and misses the metaphorical meaning entirely. In Figure 3 the "End2End" baseline exemplifies this, failing to grasp the subversion of the fairy tale trope.

**2. Over-Inference**: This happens when a model incorrectly applies a known symbol or narrative without considering the full context. The "CoT" baseline in Figure 3 demonstrates this by connecting the heart symbol to a traditional fairy tale transformation without recognizing the comic's twist.

LAD succeeds by first creating a more structured understanding in the Perception stage and then grounding its reasoning with targeted external knowledge from the Search stage, which helps avoid both superficiality and incorrect inferences.

### F.2  Analysis of Model Scaling and Image Implication Types

Our experiments have some insightful findings:

**1. Model Scaling**: By testing on QwenVL-2.5-7B and QwenVL-2.5-72B, we can analyze the effect of model scale. Our findings align with expectations: larger parameter models generally achieve better baseline performance, and both scales benefit from the LAD framework. This confirms that our method is effective across different model sizes.

**2. Image Implication Types**: Our benchmark was already designed to be diverse across various domains (e.g., life, society, art, psychology, Chinese traditional culture) and image types (e.g., comic, poster, meme). We find that models perform worse in domains containing abstract and complex information, like Art and Psychology. And models only observe the surface-level information and lack sufficient understanding of Chinese culture. In a further analysis using the annotations from the original II-Bench and CII-Bench, we observed that providing explicit labels for Emotion, Domain, and Rhetoric significantly enhances model accuracy, with Emotion labels providing the largest boost. This confirms that our framework's focus on identifying these elements in the Perception stage is well-founded.

## G  Limitation and Future Work

While our work represents a huge step towards image implication tasks, the LAD framework still suffers from the following limitations:

1) The search stage, particularly the websearch and multiple model calls, will make latency in generating image implications, although this is a trade-off for comprehensive knowledge retrieval. Based on our experiments, a single search question takes approximately 35s to 55s and whole search stage takes 3 mins to 5 mins to process through the entire pipeline.

2) Furthermore, although our Open-Style Question (OSQ) evaluation incorporates average multiple model calls and human consistency checks (the human-model scoring consistency reached 95.7% with 16 PhD students and researchers) to mitigate subjectivity, its foundation on the GPT-4o model judgments may still retain a degree of inherent bias.

In future work, we aim to prioritize optimizing the search strategy to enhance efficiency and reduce model calls without compromising performance, alongside further refining our evaluation method.

# H PROMPTS

In experiments, the prompts of different settings are as follows:

## H.1 EVALUATION

| Evaluation Metric | Evaluation Standard |
|---|---|
| **1. Surface-level Information:**
• Identification of primary entities within the image
• Analysis of color composition and application
• Recognition of intricate details and their significance

**2. Emotional Expression:**
• Identification of conveyed emotions (e.g., tranquility, intensity, melancholy)
• Depth of emotional resonance and its alignment with the image's theme
• Consistency of emotional expression across the image's elements

**3. Domain and Context:**
• Recognition of the image's domain (e.g., art, commerce, social commentary)
• Contextualization within its cultural, historical, or societal background
• Evaluation of the image's innovation within its domain

**4. Rhetorical Skills:**
• Identification of rhetorical devices (e.g., symbolism, contrast, personification)
• Analysis of how rhetorical techniques enhance the image's expression
• Integration of rhetorical devices with metaphorical implications to create a cohesive interpretation

**5. Deep Implications:**
• Recognition of metaphorical elements and their layered meanings
• Depth of interpretation of philosophical, cultural, or social values embedded in the image
• Evaluation of the originality and creativity in metaphorical interpretation | **[1 point]:**
Fails to capture key elements within the image (such as text, and important entities). Does not identify emotions, domain, or rhetorical devices. Only provides a superficial description of surface-level information, lacking depth and creativity, with a significant gap from the standard answer.

**[2 points]:**
Captures some key elements within the image, but the identification of emotions, domain, and rhetorical devices is vague. The description of surface-level information is relatively complete, but there is a clear deficiency in exploring deeper meanings, showing a noticeable gap from the standard answer.

**[3 points]:**
Effectively captures key elements within the image and initially identifies emotions, domain, and rhetorical devices. The description of surface-level information is relatively accurate, and there is some relevant expression of deep meanings. However, there is still room for improvement in depth and creativity, and it is generally close to the standard answer.

**[4 points]:**
Accurately captures key elements within the image and clearly identifies emotions, domain, and rhetorical devices. The description of surface-level information is detailed and precise, with a relatively deep exploration of deep meanings, demonstrating a certain level of creativity and depth. It is largely consistent with the standard answer but may have minor deficiencies in some details or depth.

**[5 points]:**
Accurately and precisely captures key elements within the image and profoundly identifies emotions, domain, and rhetorical devices. The description of surface-level information is comprehensive and precise, with unique insights into deep meanings, skillfully integrating image elements with metaphorical implications. It demonstrates exceptional creativity and depth, is highly consistent with the standard answer, and shows a profound grasp of metaphor creation and cultural understanding. |

Figure 4: Evaluation metric and evaluation standard of Open-Style Question.

**# Role**
You are an impartial judge who is familiar with Internet culture and memes, and is good at digging out and analyzing the deep meaning of Internet memes.

**## Attention**
You are responsible for evaluating the quality of the answer provided by the model for Internet culture and memes. Your evaluation should refer to the human answer and image, and score based on the Evaluation Standard.

**## Evaluation Standard**
**- [1 point]:**
Fails to capture key elements within the image (such as text, and important entities). Does not identify emotions, domain, or rhetorical devices. Only provides a superficial description of surface-level information, lacking depth and creativity, with a significant gap from the standard answer.
**- [2 points]:**
Captures some key elements within the image, but the identification of emotions, domain, and rhetorical devices is vague. The description of surface-level information is relatively complete, but there is a clear deficiency in exploring deeper meanings, showing a noticeable gap from the standard answer.
**- [3 points]:**
Effectively captures key elements within the image and initially identifies emotions, domain, and rhetorical devices. The description of surface-level information is relatively accurate, and there is some relevant expression of deep meanings. However, there is still room for improvement in depth and creativity, and it is generally close to the standard answer.
**- [4 points]:**
Accurately captures key elements within the image and clearly identifies emotions, domain, and rhetorical devices. The description of surface-level information is detailed and precise, with a relatively deep exploration of deep meanings, demonstrating a certain level of creativity and depth. It is largely consistent with the standard answer but may have minor deficiencies in some details or depth.
**- [5 points]:**
Accurately and precisely captures key elements within the image and profoundly identifies emotions, domain, and rhetorical devices. The description of surface-level information is comprehensive and precise, with unique insights into deep meanings, skillfully integrating image elements with metaphorical implications. It demonstrates exceptional creativity and depth, is highly consistent with the standard answer, and shows a profound grasp of metaphor creation and cultural understanding.

**## Standrad Answer:**
Human answer: {}

**## Constraints**
- Avoid any position biases and be as objective as possible
- Do not allow the length of the descriptions to influence your evaluation
- Output your final verdict by strictly following this format: "[ratings]"

**## Solve:**
Model answer: {}

Figure 5: The evaluation prompt of Open-Style Question (OSQ).

## H.2 END2END

| Prompt in Chinese | Prompt in English |
|---|---|
| 请根据提供的图片尝试回答以下单选题。直接回答正确选项，不要包含额外的解释。
请使用以下格式：“答案：$LETTER”，其中$LETTER是你认为正确答案的字母。

单选题：{}
答案： | Please try to answer the following multiple-choice questions based on the provided image. Answer the correct option directly without additional explanation.
Please use the following format: "Answer: $LETTER", where $LETTER is the letter of the correct answer you think.

Multiple-choice questions: {}
Answer: |

Figure 6: The end2end prompt of Multiple-Choice Question (MCQ).

| Prompt in Chinese | Prompt in English |
|---|---|
| 请结合以上图片，尽可能分析理解图片的深层含义。无需描述图片，仅回答图片隐喻。请保证回答的准确性并尽量简洁。 | Please try to understand the deep meaning of the image.
No need to describe images and text, only answer metaphors. Ensure the accuracy of the answer and try to be concise as much as possible. |

Figure 7: The end2end prompt of Open-Style Question (OSQ).

## H.3 COT

| Prompt in Chinese | Prompt in English |
|---|---|
| 请根据提供的图片尝试回答以下单选题。逐步思考回答正确选项，不要包含额外的解释。
请使用以下格式：“答案：$LETTER”，其中$LETTER是你认为正确答案的字母。

单选题：{}
答案： | Please try to answer the following multiple-choice questions based on the provided image. Let's think step by step to answer the correct option directly without additional explanation.
Please use the following format: "Answer: $LETTER", where $LETTER is the letter of the correct answer you think.

Multiple-choice questions: {}
Answer: |

Figure 8: The CoT prompt of Multiple-Choice Question (MCQ).

| Prompt in Chinese | Prompt in English |
|---|---|
| 请结合以上图片，逐步思考尽可能分析理解图片的深层含义。无需描述图片，仅回答图片隐喻。请保证回答的准确性并尽量简洁。 | Please try to think step by step to understand the deep meaning of the image.
No need to describe images and text, only answer metaphors. Ensure the accuracy of the answer and try to be concise as much as possible. |

Figure 9: The CoT prompt of Open-Style Question (OSQ).