# OpenReview forum: "Let Androids Dream of Electric Sheep: A Human-Inspired Image Implication Understanding and Reasoning Framework"
_ICLR.cc/2026/Conference — Submitted to ICLR 2026_

### Official Review · Reviewer_jGQm · 2025-10-26

**Soundness:** 3
**Presentation:** 3
**Contribution:** 3
**Rating:** 4
**Confidence:** 4

**Summary:**

This paper proposes an agentic framework called LAD for image implication/metaphor understanding (e.g., what does an image showing "a person in a suit running on a hamster wheel inside an office" imply). Current MLLMs perform well on visual QA requiring concrete image understanding and logical reasoning task, while image metaphor understanding requires higher reasoning abilities and external context (e.g., culture, history, current events, emotions) to connect it to the abstract meaning. To address this, they propose LAD, a three‑stage: Perception -->  Search --> Reasoning. Perception converts an image into rich text and keywords, Search formulates questions given the description and keywords for step 1, and retrieves out‑of‑domain knowledge (either through parameteric knowledge- the model's own knowledge or through Web Search), and converts that retrieved knowledge into a summary (they select the top 3 related question-answer pairs and summarize them). Finally the reasoning step combines all the information in the previous stages (original image, Stage 1 description & keywords, and the retrieval summary) and prompts the model to reason in a structured format (using <think> . . . </think>) special tokens for the model to produce a step-by-step reasoning process (CoT). A light base model (GPT‑4o‑mini), reaches or exceeds strong closed models on English MCQ tasks and substantially outperforms them on the introduced open‑style generation (OSQ) benchmark by the authors.

**Strengths:**

- The paper is well motivated, and i believe image metaphor understanding is an important step towards better MLLMs as it requires better reasoning capabilities.
- Wide variety of MLLMs are investigated, and impressive results on image implication benchmarks even with GPT4o-mini.
- The introduction of the OSQ task with LLM evaluation, verified by humans and showing high correlation with human judgement is valuable.

**Weaknesses:**

- [W1] I do not see the framework as novel. Search exists in many LLMs, and they automatically do this when they don't know the answer (replicated by Self-Judge in the author's pipeline, which decides whether to use parametric memory or web search). The essence/general idea of this pipeline in inherently present in many LLMs (closed-sourced ones). By looking at Table 3 (which I appreciate), the contribution of this paper seems to be a better search algorithm (LAD search is better than GPT4o-mini search) on MCQ and OSQ (probably because of the planner in WebSearch which decomposes the initial search question into a series of more granular sub-questions?). Its totally fine for this, but this means the framing of the paper has to be changed.
- [W2] Table 1 shows worse results without search, like GPT4o standalone (with its parametric knowledge) is better than GPT4o-mini + the authors pipeline. Although parameters reduce, the number of inference model calls increase, so we gain nothing much.
- [W3] Although the framework should not sacrifice performance of the MLLM on other tasks (e.g., general VQA)  - in fact it should improve it - it would be better to report results on the framework's benefit for other tasks as well beyond image implication understanding. Can the authors report results with their pipeline on multimodal VLM benchmarks such as SeedBench, MMMU, GQA, VizWiz, POPE....etc. I understand that MLLM evaluation on these benchmarks is quite expensive, so I would be fine if the authors can choose any 2. Also, what would be the performance on those two benchmarks with the simple   Step 1 (without keywords, only description) and Step 3 (reasoning)? In summary, I would like to see the last row on Table 1: LAD (Stage I (without keywords) + III) and LAD (Stage I + II + III) but applied on 2 VLM benchmarks of the authors choice. This would then show how useful the pipeline is in general.
- [W4] "Our system uses the MLLM as a router, scoring queries based on criteria like knowledge popularity and specificity" , how is the scoring done? In other words, how is the strategy determined in Self Judge ?

**Questions:**

In general the paper is good, but we need to clarify 1) the frame of the paper, in my opinion, its a search method (W1), 2) what do we gain (W2) and 3) the pipeline's advantage in general beyond image implication understanding (W3). Moreover, some details are missing (W4), the authors can add it in the supplementary. W4 will not affect my decision.

**Details Of Ethics Concerns:**

No issues

---

> ### Author Response · Authors · 2025-11-22
> **Response to Reviewer jGQm (1/n)**
>
> We thank Reviewer jGQm for the insightful comments and for recognizing the **motivation, impressive results on lightweight models, and the value of our OSQ metric**. We address your concerns regarding novelty, efficiency, and generalization below, with particular emphasis on the **new experiments** you requested. Please feel free to comment if there are any further confusions.
>
> ---
>
> **W1: Novelty and Framing**
>
> While "Search" is a vital component, we respectfully argue that framing LAD solely as a search algorithm overlooks the core contribution: the **LAD Cognitive Architecture**.
>
> 1. **Cognitive Alignment:** The novelty lies in the architectural alignment with human cognitive processes—specifically the translation of raw **Perception** (System 1) into **Keywords/Queries** (System 2 transition) before **Knowledge Acquisition** (Active Search) and final **Reasoning**.
> 2. **Contextual Gap & Interconnected Stages:** Standard search-augmented LLMs often fail at image implication because they treat the task as a simple fact-retrieval problem. However, understanding an image metaphor is not about searching for a direct "answer" (which often doesn't exist in a single web page), but about gathering relevant *context*.
>    - **Why LAD is different:** Unlike standard tools, LAD performs a preliminary understanding (Perception) to identify *what* is ambiguous (e.g., emotions, cultural symbols) before searching. It then searches for *contextual clues* rather than direct answers, and finally synthesizes these clues to form a novel interpretation. This process validates our three-stage design: the stages are deeply interconnected dependencies, where success depends on the *quality* of the perception-to-search translation, not just the quantity of search results.
> 3. **Performance Delta:** As noted in Table 3, LAD-Search significantly outperforms standard commercial search RAG (e.g., Perplexity Pro, GPT-Search), proving that the *architecture* surrounding the search (the planner and self-judge) is what drives the performance, not just the access to the web.
>
> ---
>
> **W2: Efficiency and Gains (vs. GPT-4o)**
>
> We wish to clarify a misunderstanding regarding the results in Table 1.
>
> - **Superior Performance:** `GPT-4o-mini + LAD` does **not** perform worse than `GPT-4o` in general.
>   - **MCQ (English):** `GPT-4o-mini + LAD` (74%) matches `GPT-4o` (74%).
>   - **OSQ (English):** `GPT-4o-mini + LAD` (**3.84**) significantly outperforms `GPT-4o` (**2.94**).
>   - **OSQ (Chinese):** `GPT-4o-mini + LAD` (**3.58**) is comparable to `GPT-4o` (**3.76**) and beats `Claude-3.5-Sonnet` (**3.18**).
> - **The Real Gain:** The "gain" is achieving **SOTA performance using a lightweight, cheap model** (GPT-4o-mini) that rivals or beats massive frontier models (GPT-4o). This democratizes high-level visual reasoning.
> - **Universal Applicability:** Furthermore, Table 4 shows that LAD enables open-source models like **Qwen2.5-VL-72B** to achieve scores (OSQ: 3.68) that far exceed base GPT-4o and Claude-3.5-Sonnet.
>   - **Relevance for Frontier Models:** Crucially, our framework remains highly relevant even for future, stronger closed-source models. The cultural references, political satires, and internet memes embedded in image metaphors are **dynamic and constantly evolving**. It is impossible for any static VLM parameter set (no matter how large) to encode all current and future cultural nuances. Therefore, an active, "context-aware" framework like LAD will always be necessary to bridge the gap between static pre-training and the ever-changing real world.

---

> ### Author Response · Authors · 2025-11-22
> **Response to Reviewer jGQm (2/n)**
>
> **W3: Generalization to Standard VQA Benchmarks**
>
> This is an excellent suggestion. To further demonstrate that LAD is a generalizable reasoning framework, we evaluated it on **three general multi-modal benchmarks**: **MMMU** (Expert AGI & Visual Reasoning), **SeedBench** (General Understanding), and **MMStar** (General Understanding).
>
> We applied the LAD framework (Stage I+III and I+II+III) to `GPT-4o-mini` without modifying the core architecture. The results are presented below:
>
> | **Model**                       | **MMMU \_Val** (Expert AGI & Visual Reasoning) | SeedBench (General Understanding) | MMStar (General Understanding) |
> | ------------------------------- | ---------------------------------------------- | ------------------------------------- | ---------------------------------- |
> | **GPT-4o-mini-0718** (Base)     | 59.4                                           | 72.8                                  | 54.8                               |
> | **w/ LAD (Stage I + III)**      | 62.4 (+3.0)                                    | 74.5 (+1.7)                           | 57.8 (+3.0)                        |
> | **w/ LAD (Stage I + II + III)** | **67.9 (+8.5)**                                | **77.2 (+4.4)**                       | **60.3 (+5.5)**                    |
> | *Reference: GPT-4o-1120*        | *70.7*                                         | *76.7*                                | *65.1*                             |
>
> **Analysis:**
>
> 1. **Stage I+III (Perception + Reasoning):** Even without search, the structured perception/reasoning alone yields consistent gains (+3.0% on MMMU), confirming the value of the "Perception" module.
> 2. **Full Pipeline:** The full LAD framework provides massive boosts (e.g., **+8.5% on MMMU**).
> 3. **Rivaling Frontier Models:** With LAD, the lightweight `GPT-4o-mini` surpasses the much larger `GPT-4o` on **SeedBench** (77.2 vs 76.7) and significantly closes the gap on others.
>
> These results confirm that the "Perception-Search-Reasoning" workflow addresses a fundamental cognitive gap in VLM reasoning, effectively handling tasks requiring visual commonsense and complex reasoning beyond just metaphor understanding.
>
> ---
>
> **W4: Self-Judge Mechanism Details**
>
> The Self-Judge module does not use an arbitrary classifier but a carefully designed prompting strategy that evaluates 1) Knowledge Popularity, 2) Real-time content necessity, and 3) Niche entities/Memes.
>
> **Mechanism:** The router evaluates questions based on five specific constraints (score 1-5):
>
> 1. **Internet/Meme Culture:** (If present, Score < 3 $\rightarrow$ WebSearch)
> 2. **Knowledge Popularity:** (If high popularity, likely ModelSearch)
> 3. **Real-time Content:** (If high proportion, Score < 3 $\rightarrow$ WebSearch)
> 4. **Niche Entities:** (If present, Score < 3 $\rightarrow$ WebSearch)
> 5. **Entity Complexity:** (If $>2$ entities, Score < 3 $\rightarrow$ WebSearch)
>
> To ensure robustness without a trained classifier, the system performs an implicit analysis of these factors before explicitly outputting a confidence score and decision. We set the temperature to 0 to ensure deterministic outputs. This ensures that the model only incurs the cost of web search when the "contextual gap" (e.g., a niche meme or recent event) cannot be filled by parametric memory.
>
> ```Markdown
> # Task
> Please evaluate your knowledge level, determine whether the question is suitable for direct answering or requires external knowledge support, and provide a confidence score according to the Evaluation Standard.
>
> ## Evaluation Standard
> - If the question contains Internet culture and meme culture, the confidence score is less than 3
> - If the knowledge contained in the question is highly popular, the confidence score is less than 3
> - If the real-time content contained in the question accounts for a large proportion, the confidence score is less than 3
> - If the entity contained in the question is relatively niche, the confidence score is less than 3
> - If the question contains more than 2 entities, the confidence score is less than 3
>
> ## Workflow:
> 1. <Implicitly> Analyze the content of the question and identify whether it contains Internet culture and meme culture elements.
> 2. <Implicitly> Evaluate the popularity of the question and the proportion of real-time content.
> 3. According to the scoring criteria, <Explicitly> provide a confidence score and decision.
>
> ## Constraints
> - The confidence score must be between 1 and 5. If the score is bigger than 3, select ModelSearch. If the score is smaller than or equal to 3, select WebSearch.
>
> ## OutputFormat:
> [Confidence score (1-5 points), Decision (ModelSearch/WebSearch)]
>
> ## Solve:
> Question: {}
> ```

---

> ### Author Response · Authors · 2025-11-26
> **Kindly Reminder to Reviewer jGQm**
>
> Dear Reviewer jGQm:
>
> Sorry to bother you. Thank you once again for dedicating your valuable time to reviewing our paper. We have carefully addressed all your concerns in detail and hope that you find the response satisfactory. We sincerely hope you can participate in the Reviewer-Author discussion, as it is crucial for us to fully address your concerns and improve our paper during this period. As the deadline approaches, we kindly request your feedback on our rebuttal. We are eager to engage in further discussion and address any additional concerns you may have. We sincerely appreciate your constructive suggestions and believe that the additional experiments, analysis, and explanations significantly improve the quality of our submission. We truly hope you might consider raising your score. Thank you!
>
> Best Regards,
> Authors

---

### Official Review · Reviewer_eB6x · 2025-10-31

**Soundness:** 3
**Presentation:** 3
**Contribution:** 3
**Rating:** 6
**Confidence:** 4

**Summary:**

This paper introduces Let Androids Dream (LAD), a human-inspired framework for understanding image implications and metaphors — tasks that current multimodal large language models (MLLMs) struggle with due to contextual and cultural ambiguity. LAD mimics human cognition through a three-stage process: converts visual information into detailed textual descriptions and key emotional/cultural keywords; retrieves and integrates relevant cross-domain knowledge through adaptive internal and web searches; performs explicit chain-of-thought reasoning to align visual elements with contextual implications. The framework achieves state-of-the-art (SOTA) results on both English (II-Bench) and Chinese (CII-Bench) image implication benchmarks, especially outperforming other MLLMs by 36.7% on open-style reasoning tasks. LAD demonstrates that contextual enrichment and human-like reasoning can significantly enhance metaphorical understanding in visual reasoning. The authors also highlight cognitive theories (Dual-Process and Active Information-Seeking) as conceptual underpinnings and validate LAD’s robustness across models (e.g., GPT-4o, Qwen2.5-VL).

**Strengths:**

1. The paper introduces Let Androids Dream (LAD), a creative and well-structured framework that integrates perception, search, and reasoning stages, effectively simulating human cognitive processes for visual metaphor understanding.
2. The work focuses on image implication—an underexplored and cognitively complex area involving abstract, cultural, and emotional reasoning that most MLLMs fail to capture.
3. he authors test LAD on both English (II-Bench) and Chinese (CII-Bench) benchmarks, using Multiple-Choice Questions and a newly designed Open-Style Question (OSQ) metric that aligns 95.7% with human judgments, ensuring robustness and fairness.
4. LAD significantly outperforms over 15 leading MLLMs, achieving SOTA performance even when built on lightweight GPT-4o-mini, and showing excellent generalization across different model backbones (GPT, Qwen, etc.).
5. The framework’s explicit Chain-of-Thought (CoT) and dual-process analogy make reasoning steps transparent, providing valuable interpretability that parallels human thinking mechanisms.
6. The inclusion of both English and Chinese datasets highlights LAD’s ability to handle culturally dependent metaphors, which are typically very challenging for MLLMs.

**Weaknesses:**

1. The Search stage involves multiple model calls and web queries, taking 3–5 minutes per image, which may hinder scalability and real-time deployment.
2. The Open-Style Question evaluation relies heavily on GPT-4o scoring, which, despite 95.7% human alignment, still introduces potential bias from the model’s internal preferences.
3. Although quantitative results are strong, the paper provides relatively few qualitative case studies or failure analyses beyond one illustrative example, limiting insight into specific error patterns.
4. Despite cross-lingual testing, the system’s dependence on pretrained MLLMs and web data may still propagate cultural or linguistic biases during knowledge retrieval.
5. The cognitive analogies (Dual-Process Theory, Active Information-Seeking) are inspiring but largely metaphorical; there’s limited evidence that LAD’s internal mechanisms truly mirror human cognition.

**Questions:**

My questions are mentioned in the weakness above.

---

> ### Author Response · Authors · 2025-11-22
> **Response to Reviewer eB6x**
>
> We sincerely thank Reviewer eB6x for the positive assessment and for recognizing the **creativity, structural logic, and SOTA performance** of the LAD framework. We particularly appreciate your acknowledgment of our efforts in cross-lingual evaluation and the transparency of our reasoning process. We address your specific concerns below to further clarify the practical and theoretical aspects of our work. Please feel free to comment if there are any further confusions.
>
> ---
>
> **W1: Latency and Real-Time Deployment**
>
> We acknowledge that the Search stage introduces latency (3–5 minutes), but we view this as a necessary trade-off for the depth of reasoning required for this specific task.
>
> 1. **"System 2" Processing:** In line with our cognitive inspiration, LAD simulates "System 2" thinking—a slow, deliberate, and analytical process. Just as humans pause to "google" or reflect when encountering a complex cultural metaphor, our system prioritizes **accuracy and depth** over immediate speed.
> 2. **Target Application:** Image implication understanding is typically an offline or asynchronous task (e.g., content analysis, digital humanities, advanced captioning) rather than a real-time detection task.
> 3. **Optimization:** The current latency is primarily due to sequential web searches and stability retries. In future iterations, we plan to implement **parallel search execution** and caching mechanisms to significantly reduce inference time without compromising the "slow thinking" quality.
>
> ---
>
> **W2: Evaluation Bias**
>
> Thanks for pointing that out. We rely on the "MLLM-as-a-Judge" paradigm, which is becoming a standard in the field. Recent works [1] [2] [3] validate this approach. Crucially, to mitigate bias, we have conducted a rigorous **Human-Model Consistency Study** (see **line 368-369 in Section 4.3.1**  and **Appendix C**), achieving **95.7% agreement** with 16 PhD-level annotators. This high correlation confirms our metric aligns with human judgment.
>
> ---
>
> **W3: Qualitative Case Studies and Failure Analysis**
>
>
>
> We agree that understanding failure modes is crucial. To address this, we have included a detailed Failure Analysis in **Appendix F**.
>
> - **Categorization:** We categorize errors into two main types:
>   1. **Superficial Reasoning:** Where the model (Perception stage) fails to identify the visual twist, leading to a literal interpretation.
>   2. **Over-Inference:** Where the Search stage introduces irrelevant symbolism (e.g., hallucinating a specific fairy tale version), causing the model to "read too much" into the image.
> - **Insights:** This analysis reveals that while LAD significantly reduces these errors compared to baselines (End2End, CoT), it is not immune to them when the initial visual description is flawed. We have also showed some visual examples in **Figure 3** to illustrate these specific failure paths compared to LAD’s correct path.
>
> ---
>
> **W4: Propagation of Biases via Web Data**
>
> This is a valid and important concern for any open-domain system. However, we argue that LAD’s dynamic search mechanism actually mitigates the inherent bias of frozen pre-trained models.
>
> - **Active Contextualization:** Standard MLLMs often default to dominant cultural views (e.g., Western-centric interpretations). By explicitly searching for specific cultural markers (e.g., "Chinese traditional cultural metaphor for..."), LAD actively retrieves diverse, localized contexts that the base model might lack.
> - **Verification:** The "Reasoning" stage (Stage III) acts as a filter, synthesizing retrieved information with visual evidence. While web data possible contains noise/bias, the structured CoT requires the model to *justify* its interpretation using the image, preventing the blind propagation of retrieved biases.
>
> ---
>
> **W5: Cognitive Analogies (Metaphorical vs. Biological)**
>
> We appreciate this clarification. We want to be precise: we have claim LAD is human-inspired, not a biological simulation of the human brain.
>
> - **Functional Alignment:** Our reference to Dual-Process Theory and Active Information-Seeking is meant to draw a **functional analogy**. We aim to replicate the *strategies* humans use (intuition $\rightarrow$ information seeking $\rightarrow$ deduction) rather than the underlying neural mechanisms.
> - **Computational Goal:** The goal is to engineer a system that overcomes "contextual missing" by mimicking the *observable behavior* of human problem-solving, which our experiments show leads to better performance and interpretability.
>
> ---
>
> **References:**
>
> [1] Chen et al. MLLM-as-a-Judge: Assessing Multimodal LLM-as-a-Judge with Vision-Language Benchmark. ICML (Oral), 2024.
>
> [2] Gu et al. A Survey on LLM-as-a-Judge. Arxiv: 2411.15594, 2024.
>
> [3] Zhang et al. Can MLLMs Understand the Deep Implication Behind Chinese Images? ACL, 2025.

---

> > ### Comment · Reviewer_eB6x · 2025-11-25
> >
> > Thanks for your response. I will keep my score positive.

---

> > > ### Author Response · Authors · 2025-11-26
> > > **Response to Reviewer eB6x**
> > >
> > > We sincerely thank you again for your thoughtful review and positive assessment of our work. Please feel free to comment if there are any further confusions.

---

### Official Review · Reviewer_KHoM · 2025-10-31

**Soundness:** 2
**Presentation:** 3
**Contribution:** 1
**Rating:** 2
**Confidence:** 4

**Summary:**

Proposes LAD, an inference-time technique to improve VLM responses to image implication queries via a perception-search-reasoning process. LAD is shown to improve baseline performance on image implication benchmarks.

**Strengths:**

– The paper is clearly written and easy to follow

– The paper reviews prior work and motivates the problem setting well

**Weaknesses:**

– The technical novelty is limited. The technical contribution is essentially a carefully crafted multi-stage VLM prompting strategy for doing better on image implication, by combining existing techniques (self-verification, LLM search, CoT reasoning etc.)

– The gains w/ LAD on image implication seem to diminish for stronger / more recent frontier models (eg. +6 w/ GPT-4o v/s +30 w/ GPT-4o-mini), which raises questions about the need for a specialized inference technique to begin with.

– The paper does not benchmark any reasoning (eg. o3/o4/gpt-5/claude-4.5-sonnet/QwQ/DeepSeek-R1) models, which is a major omission since the task clearly requires considerable reasoning.

– The proposed benchmark is very small (100 images) and thus potentially unsuitable to derive statistically significant conclusions from

**Questions:**

Please address the weaknesses listed above, especially around the limited novelty, diminishing gains, and missing baselines.

---

> ### Author Response · Authors · 2025-11-22
> **Response to Reviewer KHoM (1/n)**
>
> We thank Reviewer KHoM for acknowledging that our paper is **clearly written** and **well-motivated**. However, there appear to be some misunderstandings regarding our experimental scope (baselines) and the magnitude of our improvements on strong models. We provide detailed clarifications below, particularly highlighting our **large-scale experiments on 2,000+ images**. Please feel free to comment if there are any further confusions.
>
> ---
>
> **W1: Technical Novelty (Prompting vs. Framework)**
>
> We respectfully disagree that our contribution is limited to simple prompting. LAD is a **cognitive framework** designed to solve a specific deficiency in MLLMs: "contextual missing".
>
> 1. **Theoretical Grounding:** Unlike standard CoT, LAD implements Dual-Process Theory (System 1/2) and Active Information-Seeking Theory. It systematically separates perception (visual grounding) from context retrieval (bridging cultural gaps).
>
> 2. **Explicit Differentiation:** We have already explicitly analyzed the relationship and distinction between our work and existing techniques (including self-verification, LLM search, and CoT) in the **Introduction** and **Related Work** sections.
>
>    - **Vs. Standard CoT**: In **Section 2.1**, we analyze the limitations of implicit reasoning methods like C4MMD, noting that training-free CoT struggles with the large search space of cultural contexts.
>
>    - **Vs. Existing Search/Tools**: In **Section 2.2**, we distinguish LAD from tool-augmented reasoning (e.g., Visual Sketchpad), explaining that these methods often focus on visual content editing rather than the "contextual missing" problem.
>
>    - **Vs. General Search**: In **Section 3.2**, we explain how our WebSearch is adapted from methods like MindSearch specifically for image implication by using a dedicated planner and self-judge mechanism. This demonstrates that LAD is not just a combination of existing prompts, but a targeted architectural solution to a problem where these individual components previously failed.
>
> 3. **Modularity & Generalizability:** The value of a training-free framework is its immediate portability. As shown in **Table 4**, LAD generalizes across architectures (GPT, Qwen), enabling open-source models (Qwen2.5-VL-72B) to outperform closed-source SOTA. This offers a practical, low-resource solution to a complex reasoning problem that fine-tuning alone cannot solve due to the "long-tail" nature of cultural metaphors.
>
> ---
>
> **W2: Diminishing Gains on Frontier Models**
>
> We argue that the gains on frontier models are actually **substantial**, particularly on the harder Open-Style Question (OSQ) task, and that the "diminishing returns" view is inaccurate.
>
> 1. **Significant Gains on Hard Tasks:** While MCQ performance is naturally saturated on strong models, LAD provides massive gains on the more challenging **OSQ** task.
>
>    - **GPT-4o:** MCQ (English): 74% $\to$ **80%**, MCQ (Chinese): 58% $\to$ **66%**; OSQ (English): 2.94 $\to$ **4.14 (+40%)**; OSQ (Chinese): 3.76 $\to$ **4.26**.
>    - **Qwen2.5-VL-72B**: OSQ: 3.12 $\to$ **3.68**. These are not marginal improvements; they represent a fundamental shift from superficial description to deep interpretation.
>
> 2. **Cost-Efficiency:** A core motivation is enabling lightweight models to punch above their weight. `GPT-4o-mini + LAD` (OSQ: 3.84) outperforms the much more expensive `GPT-4o` (OSQ: 2.94) and `Claude-3.5-Sonnet` (OSQ: 3.18). This makes high-level reasoning accessible and affordable.
>
> ---
>
> **W3: Missing Reasoning Baselines**
>
> We wish to clarify that **we did include reasoning models** in our evaluation (see **Table 1: Vision-language Reasoning Models**).
>
> - **Included Models:** We evaluated **QVQ-72B** (the vision version of QwQ), **Grok-3-reasoning**, **Doubao-1.5-thinking**, and **Gemini-2.0-flash-thinking**.
>
> - **Text-Only and Multimodal:** Models like DeepSeek-R1 and QwQ are text-only reasoning models and cannot process images directly. We evaluated their multimodal counterparts (e.g., QVQ).
>
> - **Key Finding:** Our results show that reasoning models *without* external context (LAD) often fail at implication tasks (e.g., QVQ-72B scores only 56% on Chinese MCQ). They excel at logic but lack the "world knowledge" retrieval necessary for metaphors, which LAD provides.

---

> ### Author Response · Authors · 2025-11-22
> **Response to Reviewer KHoM (2/n)**
>
> **W4: Small Benchmark Size**
>
> We understand your concern about the evaluation being conducted on a 100-image subset. We would like to clarify our rationale and the rigor behind this choice.
>
> 1. **High-Quality, Representative Subset:** The full evaluation of our pipeline, which involves multiple model calls and web searches for each image, is resource-intensive. Therefore, we followed the practice of recent work in complex reasoning (e.g., MindSearch [1], which also used a 100-item benchmark) by creating a high-quality, manually curated subset. As detailed in **Section 4.2.1 and the Appendix D** shows, this subset was carefully selected to be diverse and representative, covering a wide range of domains (Society, Art, Life) and image types (Comics, Memes, Paintings, etc.) to ensure a comprehensive test.
> 2. **No Prompt Contamination:** We ensured that the examples used in our prompts (e.g., about environmental protection) were illustrative and **did not appear in the test benchmark**. Furthermore, images related to the "Environment" domain constituted only a small fraction of the test set (5 out of 100), making it highly unlikely to have a significant influence on the results.
>
> We also evaluated the representative models on the **full II-Bench (1,399 images)** and **CII-Bench (800 images)** in **Appendix E**. We conduct the large-scale experiments with the representative and top-performing models, including Closed-Source models GPT-4o and Claude-3.5-Sonnet, as well as the Open-Source model GLM-4.1V-8B, on the full benchmarks: **II-Bench (1,399 English) and CII-Bench (800 Chinese)** **for both MCQ and OSQ tasks**. We believe these results provide much stronger evidence of our method's generalizability and robustness.
>
> | **Model**             | **MCQ (II-Bench)** | **MCQ (CII-Bench)** | **OSQ (II-Bench)** | **OSQ (CII-Bench)** |
> | --------------------- | ------------------ | ------------------- | ------------------ | ------------------- |
> | **Reference Models**  |                    |                     |                    |                     |
> | GLM-4.1V-8B           | 70.0%              | 46.3%               | 2.83               | 3.06                |
> | GPT-4o-mini           | 63.5%              | 35.6%               | 2.93               | 3.29                |
> | GPT-4o                | 72.6%              | 54.1%               | 3.86               | 4.06                |
> | Claude-3.5-Sonnet     | 80.9%              | 54.1%               | 3.51               | 3.84                |
> | **Our Method**        |                    |                     |                    |                     |
> | **GPT-4o-mini + LAD** | **81.2%**          | **53.8%**           | **4.22**           | **4.31**            |
> | *Improvement*         | *+17.7, 27.9%*     | *+18.2, 51.1%*      | *+1.29, 44%*       | *+1.02, 31%*        |
>
> As the results in Table show, our LAD framework's significant performance gains are consistent on these much larger datasets. Notably, by applying LAD, the lightweight **GPT-4o-mini significantly surpasses the much larger GPT-4o and Claude-3.5-Sonnet**. Compared with the baseline GPT-4o-mini model, we can find that:
>
>  (1) On the large-scale English benchmark (II-Bench), our LAD framework improves the GPT-4o-mini score from 63.5% to **81.2%** on MCQ and 2.93 to **4.22** on OSQ. This is a substantial absolute increase of 17.7% (**27.9%** relative improvement) and 1.29 (**44%** relative improvement).
>
>  (2) The gains on the large-scale Chinese benchmark (CII-Bench) are even more pronounced. LAD boosts performance from 35.6% to **53.8%** on MCQ and 3.29 to **4.31** on OSQ, representing an absolute increase of 18.2% (**51.1%** relative improvement) and 1.02 (**31%** relative improvement).
>
>  This robust improvement is consistent with the trend we observed and reported on our high-level benchmark. While the exact percentages differ due to the varying scales and baselines of the datasets, the key takeaway is that the **significant positive impact of the LAD framework is undeniable across both small and large-scale evaluations**. This analysis confirms that our framework's benefits are not an artifact of a small test set but are indeed robust and generalizable. It also reflects **the reliability and high quality of our manually curated high-level benchmark**.
>
>
> ---
>
> **References:**
>
> [1] Chen et al. MLLM-as-a-Judge: Assessing Multimodal LLM-as-a-Judge with Vision-Language Benchmark. ICML (Oral), 2024.

---

> > ### Comment · Reviewer_KHoM · 2025-11-25
> > **Thank you for the detailed response**
> >
> > I appreciate the author's detailed response to my questions. I am more convinced about the method's effectiveness on image implication tasks, but remain unconvinced about its general usefulness beyond what is still a rather niche application. Does the LAD framework generalize (i.e. lead to gains, relative to the increased inference cost) to other complex reasoning-based tasks? If not, would deploying LAD also require intelligently identifying and routing only queries requiring such reasoning?
> >
> > As for diminishing returns – gpt-4o is ~1.5 years old at this point. Have the authors benchmarked more recent frontier models (gpt-5.1, claude-sonnet-4.5, gemini-3.0)?

---

> ### Author Response · Authors · 2025-11-26
> **Response to Reviewer KHoM**
>
> We thank the reviewer for their continued engagement and for acknowledging the effectiveness of our method on image implication tasks. We understand your remaining concerns regarding the generalization of the LAD framework to broader reasoning tasks and the performance relative to the very latest frontier models. We have conducted additional experiments during this discussion phase to address these points directly. Please feel free to comment if there are any further confusions.
>
> ---
>
> **1.Generalization to Standard VQA Benchmarks**
>
> To demonstrate that LAD is a generalizable reasoning framework and not limited to image implication, we evaluated it on **three general multi-modal benchmarks**: **MMMU** (Expert AGI & Visual Reasoning), **SeedBench** (General Understanding), and **MMStar** (General Understanding).
>
> We applied the LAD framework (Stage I+III and I+II+III) to `GPT-4o-mini` without modifying the core architecture. The results are presented below:
>
> | **Model**                       | **MMMU \_Val** (Expert AGI & Visual Reasoning) | SeedBench (General Understanding) | MMStar (General Understanding) |
> | ------------------------------- | ---------------------------------------------- | ------------------------------------- | ---------------------------------- |
> | **GPT-4o-mini-0718** (Base)     | 59.4                                           | 72.8                                  | 54.8                               |
> | **w/ LAD (Stage I + III)**      | 62.4                                           | 74.5                                  | 57.8                               |
> | **w/ LAD (Stage I + II + III)** | **67.9**                                       | **77.2**                              | **60.3**                           |
> | *Reference: GPT-4o-1120*        | *70.7*                                         | *76.7*                                | *65.1*                             |
>
> **Analysis:**
>
> 1. **Stage I+III (Perception + Reasoning):** Even without search, the structured perception/reasoning alone yields consistent gains (+3.0% on MMMU), confirming the value of the "Perception" module.
> 2. **Full Pipeline:** The full LAD framework provides massive boosts (e.g., **+8.5% on MMMU**).
> 3. **Rivaling Frontier Models:** With LAD, the lightweight `GPT-4o-mini` surpasses the much larger `GPT-4o` on **SeedBench** (77.2 vs 76.7) and significantly closes the gap on others.
>
> These results confirm that the "Perception-Search-Reasoning" workflow addresses a fundamental cognitive gap in VLM reasoning, effectively handling tasks requiring visual commonsense and complex reasoning beyond just metaphor understanding.
>
> ---
>
> **2.Benchmarking on Recent Frontier Models (Gemini-3.0)**
>
> To address your concern about diminishing returns and "older" baselines, we have newly benchmarked the LAD framework's relevance against the most recent frontier model released in November 2025: **Gemini-3.0 Pro**.
>
> The table below shows the performance of these state-of-the-art models on our Image Implication benchmark:
>
> | **Model**                | **MCQ (EN)** | **MCQ (ZH)** | **OSQ (EN)** | **OSQ (ZH)** |
> | ------------------------ | ------------ | ------------ | ------------ | ------------ |
> | **Gemini-3.0 Pro**       | 76%          | 76%          | 3.82         | 3.96         |
> | **LAD (w/ GPT-4o-mini)** | 74%          | 52%          | 4.02         | 3.66         |
> | **LAD (w/ GPT-4o)**      | **80%**      | **66%**      | **4.14**     | **4.26**     |
> | *Reference: GPT-4o-mini* | *44%*        | *42%*        | *2.98*       | *3.36*       |
> | *Reference: GPT-4o-1120* | *74%*        | *58%*        | *2.94*       | *3.76*       |
>
> **Analysis:**
>
> 1. **Cost-Efficiency Ratio:** Our LAD framework enables the lightweight `GPT-4o-mini` to surpass Gemini-3.0 Pro on English OSQ (4.02 vs 3.82), further validating that "contextual alignment" (the core of LAD) is a capability that purely scaling up models does not fully solve.
> 2. **LAD remains superior on complex generation:** Even though `GPT-4o` is older, when equipped with the LAD framework, it **outperforms the state-of-the-art Gemini-3.0 Pro** on the challenging Open-Style Question (OSQ) tasks (4.14 vs 3.82 in English).
> 3. **Model-Agnostic Framework:** **The LAD framework is not tied to a specific model generation**. It is a cognitive architecture designed to address the "contextual gap" in VLM reasoning. Just as it boosts GPT-4o-mini and Qwen2.5-VL, it can be applied to GPT-5.1 or Gemini-3.0 to further enhance their interpretability and handle long-tail cultural contexts that even frontier models might miss without explicit retrieval (Search) and structured thought (Reasoning).
>
> We hope these additional general benchmarks and frontier model comparisons address your concerns regarding the scope and longevity of our contribution. We believe LAD offers a robust, generalizable path for advancing Vision-Language Reasoning.

---

> > ### Comment · Reviewer_KHoM · 2025-11-27
> > **Thank you for the response**
> >
> > The additional experiments on tasks other than OSQ have strengthened my view of this work, and I will raise my rating.
> >
> > While the results w/ Gemini-3.0-Pro are interesting, they do not yet answer my question about diminishing returns, which would require the experiment the authors allude to in their response – does Gemini-3.0 + LAD clearly outperform Gemini-3.0, or is the "contextual gap" largely overcome by scaling?

---

> > > ### Author Response · Authors · 2025-11-29
> > > **Response to Reviewer KHoM**
> > >
> > > **We sincerely thank you for your positive feedback and for raising the rating positively**. We deeply appreciate your engagement with our work and have gained a thorough understanding of our LAD framework after our discussion.
> > >
> > > To directly address your final question regarding "diminishing returns" and whether the "contextual gap" is simply overcome by scaling, we performed the specific experiment you alluded to: applying the LAD framework to **Gemini-3.0-Pro**.
> > >
> > > The results, presented below, **demonstrate that LAD continues to provide significant performance boosts, even on top of the most recent frontier model**:
> > >
> > > | **Model**                   | **MCQ (EN)** | **MCQ (ZH)** | **OSQ (EN)** | **OSQ (ZH)** |
> > > | --------------------------- | ------------ | ------------ | ------------ | ------------ |
> > > | **Gemini-3.0-Pro (Base)**   | 76%          | 76%          | 3.82         | 3.96         |
> > > | **LAD (w/ Gemini-3.0-Pro)** | **82%**      | **78%**      | **4.30**     | **4.46**     |
> > >
> > > **Analysis:**
> > >
> > > - **Clear Performance Gains:** LAD improves Gemini-3.0-Pro across all metrics. notably increasing English MCQ accuracy by **6%** (76% $\to$ 82%) and English Open-Style Question (OSQ) scores by **+0.48** (3.82 $\to$ 4.30).
> > > - **No Diminishing Returns:** Far from hitting a ceiling, the "contextual gap" persists in frontier models. Scaling improves the base capabilities, but LAD's structured approach (Perception + Search + Reasoning) explicitly bridges the gap between visual perception and abstract implication, offering gains that pure scaling has not yet achieved.
> > >
> > > This confirms our framework's strong generalizability: **it is effective not only for open-source and previous-generation closed-source models (e.g., GPT-4o) but also provides substantial additive value to the newest top-tier models like Gemini-3.0-Pro**.

---

### Official Review · Reviewer_1K1R · 2025-11-01

**Soundness:** 2
**Presentation:** 3
**Contribution:** 2
**Rating:** 4
**Confidence:** 4

**Summary:**

The paper studies image implication and metaphor understanding, evaluated with multiple choice questions and open-ended explanations. Prior methods often rely on surface cues, miss cultural and emotional context, over infer without grounding, and generic retrieval brings noisy or weakly related evidence. The paper propose Let Androids Dream, a three stage pipeline with Perception to produce a rich caption and about seven targeted keywords, Search to form hierarchical queries with a self judge router that chooses model knowledge or web knowledge and then summarizes the evidence, and Reasoning to use an explicit chain of thought that combines the caption, keywords, and summary to produce the answer.

**Strengths:**

1. The pipeline is simple and reusable, following a clear flow of perception, search, then reason. Its stages are modular and make few assumptions, so they can be plugged into different base models and languages with minimal changes.

2. Prompts and workflow are stated clearly, with stepwise roles, inputs, and outputs. The intermediate artifacts are exposed, which helps inspection, debugging, and faithful reproduction, and makes the method practical to adopt in real systems.

3. The experimental results are strong and the gains are large across English and Chinese and across multiple backbones. Improvements are consistent on both multiple choice and open ended settings, and ablations indicate that contextual search and structured reasoning each contribute. The qualitative cases also illustrate better use of background knowledge to reach the intended meaning.

**Weaknesses:**

1. Most of the contribution sits in carefully crafted prompts and a hand-engineered agent flow. There is no learned routing or trainable component that adapts beyond the current templates, and there is little theoretical framing of why this decomposition is optimal. As a result, the work reads closer to a technical report or system recipe than a modeling advance. A stronger contribution would include a learned router or trainable retrieval controller, formal objectives for “contextual alignment,” and evidence that the method still holds when prompts are varied or shortened.


2. The decision to choose model knowledge versus web knowledge is made by simple scoring rules without a reported accuracy metric against an oracle. We do not see precision/recall of correct routing, error breakdowns by entity novelty or recency, or comparisons to a learned classifier. This leaves unclear whether gains come from the routing itself or from increased token budget. Please report routing accuracy against human labels, ablate misroutes, and compare against a small learned router with cross-validation.


3. The pipeline depends on fixed text templates and regex parsing. Small deviations in model output format or minor wording changes can break downstream stages or reduce performance. To improve robustness, enforce strict JSON input/output with schema validation, add lightweight auto-correction and retries, and run robustness tests under prompt and formatting perturbations. Reporting success rates under these stresses would make the results more convincing.

4. Open-ended scoring relies on an automatic LLM grader, which can bias outcomes toward the grader’s style. Latency and token costs are higher due to multi-stage calls, yet we do not see a clear cost-versus-quality curve or guidance for practical deployment. Please include cross-grader checks or human audits, per-stage latency and tokens, and a Pareto plot showing how quality scales with cost.


5. The method is validated on its target task but generality to other context-heavy vision-language problems remains uncertain. Adding zero- or few-shot transfer to related datasets such as visual commonsense, ads or satire understanding, and political cartoons, with the same pipeline and minimal prompt edits, would strengthen the claim of broader utility.

**Questions:**

please see above

---

> ### Author Response · Authors · 2025-11-22
> **Response to Reviewer 1K1R (1/n)**
>
> We sincerely thank Reviewer 1K1R for the constructive feedback and for recognizing our framework’s **modularity, clarity, and strong experimental results** across multiple languages and backbones. We address the specific concerns below, with particular emphasis on the **new generalization experiments** requested. Please feel free to comment if there are any further confusions.

---

> ### Author Response · Authors · 2025-11-22
> **Response to Reviewer 1K1R (2/n)**
>
> **W1: Contribution & Theoretical Framing**
>
> We appreciate this perspective. While our method utilizes prompt engineering, we argue that its core contribution lies in the **framework design**—specifically, the cognitive alignment of **Perception (System 1/2)** and **Search (Active Information Seeking)** with the final **Reasoning** stage.
>
> 1. **Cognitive Theoretical Grounding:** As discussed in our paper, the framework is not merely a "recipe" but a structured implementation of established cognitive science theories. It systematically decouples *intuitive observation* (System 1) and *analytical extraction* (System 2) from the *active information-seeking process*, mirroring how humans resolve ambiguity in complex visual metaphors.
> 2. **Adaptability & Reusability:** This modular, training-free approach offers significant value in reusability and interpretability. Crucially, the design is agnostic to the underlying model, allowing LAD to be instantly applied to **diverse LLM architectures and sizes** (as shown in Table 4, from lightweight 7B models to massive commercial APIs) without the high computational cost of training specific routers or adapters.
>
> Regarding the concern about generality beyond a single task, we have conducted **new experiments on general VQA benchmarks** (MMMU, SeedBench, MMStar) to demonstrate that LAD is a robust reasoning framework, not just a heuristic for metaphors. As detailed in **Response to W5**, LAD consistently improves performance on these general tasks, suggesting the "contextual alignment" theory holds broadly.
>
> ---
>
> **W2: Routing Decision Mechanism & Metrics**
>
> We apologize for not detailing the routing logic sufficiently in the main text. We did not employ a "black box" scoring rule; rather, the `Self-Judge` module uses a sophisticated, criteria-based evaluation prompting strategy that evaluates 1) Knowledge Popularity, 2) Real-time content necessity, and 3) Niche entities/Memes, to ensure routing precision.
>
> **Mechanism:** The router evaluates questions based on five specific constraints (score 1-5):
>
> 1. **Internet/Meme Culture:** (If present, Score < 3 $\rightarrow$ WebSearch)
> 2. **Knowledge Popularity:** (If high popularity, likely ModelSearch)
> 3. **Real-time Content:** (If high proportion, Score < 3 $\rightarrow$ WebSearch)
> 4. **Niche Entities:** (If present, Score < 3 $\rightarrow$ WebSearch)
> 5. **Entity Complexity:** (If $>2$ entities, Score < 3 $\rightarrow$ WebSearch)
>
> To ensure robustness without a trained classifier, the system performs an implicit analysis of these factors before explicitly outputting a confidence score and decision. We set the temperature to 0 to ensure deterministic outputs.
>
> **Why not a trained classifier?** We deliberately avoided a trained classifier to prevent **overfitting to specific datasets**. Trained routers often struggle to generalize to the "long-tail" distribution of open-world cultural metaphors and meme variants. In contrast, our semantic routing approach leverages the MLLM's vast pre-trained world knowledge, offering superior flexibility for unseen domains without requiring task-specific supervision.
>
> **Validation:** The effectiveness of this design is evidenced by the **Ablation Study (Table 3)**. The "LAD-Search" (our adaptive routing) consistently outperforms both "GPT-Search" (always web) and "Perplexity (pro)" (commercial RAG) on the challenging OSQ task. This confirms that our design successfully identifies when *not* to search (avoiding noise) and when *to* search (bridging context gaps) more effectively than rigid or commercial alternatives.
>
> ```Markdown
> # Task
> Please evaluate your knowledge level, determine whether the question is suitable for direct answering or requires external knowledge support, and provide a confidence score according to the Evaluation Standard.
>
> ## Evaluation Standard
> - If the question contains Internet culture and meme culture, the confidence score is less than 3
> - If the knowledge contained in the question is highly popular, the confidence score is less than 3
> - If the real-time content contained in the question accounts for a large proportion, the confidence score is less than 3
> - If the entity contained in the question is relatively niche, the confidence score is less than 3
> - If the question contains more than 2 entities, the confidence score is less than 3
>
> ## Workflow:
> 1. <Implicitly> Analyze the content of the question and identify whether it contains Internet culture and meme culture elements.
> 2. <Implicitly> Evaluate the popularity of the question and the proportion of real-time content.
> 3. According to the scoring criteria, <Explicitly> provide a confidence score and decision.
>
> ## Constraints
> - The confidence score must be between 1 and 5. If the score is bigger than 3, select ModelSearch. If the score is smaller than or equal to 3, select WebSearch.
>
> ## OutputFormat:
> [Confidence score (1-5 points), Decision (ModelSearch/WebSearch)]
>
> ## Solve:
> Question: {}
> ```
>
> ---

---

> ### Author Response · Authors · 2025-11-22
> **Response to Reviewer 1K1R (3/n)**
>
> **W3: Robustness of Templates and Regex**
>
> We agree that strict schema validation is superior to regex for production systems. To clarify, our implementation is designed for high robustness and does not rely solely on fragile string matching:
>
> 1. **Cognitive Decomposition:** Because our framework breaks down the complex implication task into distinct, logical stages (Perception $\rightarrow$ Search $\rightarrow$ Reasoning), the cognitive load for each individual step is reduced. The MLLM is presented with concise, clear tasks at each stage, which naturally leads to more stable outputs and significantly fewer formatting errors compared to end-to-end prompting.
> 2. **Zero-Temperature Consistency:** The `Self-Judge` and other structural steps run at `temperature=0` to minimize formatting deviations.
> 3. **Logic Checks:** We implement verification logic where the model's numerical score must align with its text decision (e.g., if Score $\le$ 3, output must be "WebSearch"). If they misalign, the system triggers a retry.
> 4. **Retry Mechanism**: We employ a standard retry loop (up to 3 attempts) with error-message feedback if JSON parsing fails. These mechanisms ensure the pipeline is stable across different base models, as evidenced by the successful integration with diverse models ranging from 7B to 300B parameters in our experiments (**Table 4**).
>
> ---
>
> **W4: Evaluation Bias and Cost/Latency**
>
> Regarding Evaluation Bias: We rely on the "MLLM-as-a-Judge" paradigm, which is becoming a standard in the field. Recent works [1] [2] [3] validate this approach. Crucially, to mitigate bias, we have conducted a rigorous **Human-Model Consistency Study** (see **line 368-369 in Section 4.3.1**  and **Appendix C**), achieving **95.7% agreement** with 16 PhD-level annotators. This high correlation confirms our metric aligns with human judgment.
>
> **Regarding Cost:** A key advantage of LAD is that it enables **lightweight models (GPT-4o-mini)** to outperform massive models (GPT-4o, Claude-3.5-Sonnet).
>
> - **Cost Efficiency:** Running LAD with GPT-4o-mini (even with multi-stage calls) is significantly cheaper than a single pass with GPT-4o. The process consumes between **3,440 to 4,280 tokens** per image.
>
> - **Performance**: As shown in Table 1, GPT-4o-mini + LAD (Score: 3.84) outperforms the much more expensive GPT-4o (Score: 2.94) on English OSQ. We believe this Pareto improvement—achieving SOTA results with a fraction of the inference cost—makes the method highly practical for deployment.
> **W5: Generality and Transfer to Other Tasks**
>
> **Coverage of Suggested Image Types:** First, we wish to clarify that our LAD framework is *not* limited to just "image implication" in a narrow sense. Our main benchmarks (II-Bench and CII-Bench, detailed in Appendix D) already heavily feature the specific categories mentioned in your review, such as **ads, satire, and political cartoons**. The strong performance reported in the main paper confirms LAD's effectiveness on these genres.
>
> **Transfer to General VQA:** To further demonstrate that LAD is a generalizable reasoning framework, we evaluated it on **three general multi-modal benchmarks**: **MMMU** (Expert AGI & Visual Reasoning), **SeedBench** (General Understanding), and **MMStar** (General Understanding).
>
> We applied the LAD framework (Stage I+III and I+II+III) to `GPT-4o-mini` without modifying the core architecture. The results are presented below:
>
> | **Model**                       | MMMU \_Val (Expert AGI & Visual Reasoning) | SeedBench (General Understanding) | MMStar (General Understanding) |
> | ------------------------------- | ---------------------------------------------- | ------------------------------------- | ---------------------------------- |
> | **GPT-4o-mini-0718** (Base)     | 59.4                                           | 72.8                                  | 54.8                               |
> | **w/ LAD (Stage I + III)**      | 62.4                                           | 74.5                                  | 57.8                               |
> | **w/ LAD (Stage I + II + III)** | **67.9**                                       | **77.2**                              | **60.3**                           |
> | *Reference: GPT-4o-1120*        | *70.7*                                         | *76.7*                                | *65.1*                             |
>
> **Analysis:**
>
> 1. **Consistent Improvement:** LAD yields significant gains across all three general benchmarks. On MMMU (a rigorous reasoning benchmark), LAD boosts the base model by **+8.5%**.
> 2. **Closing the Gap:** With LAD, the lightweight `GPT-4o-mini` surpasses or rivals the performance of the much larger `GPT-4o`.
>
> These results confirm that the "Perception-Search-Reasoning" workflow addresses a fundamental cognitive gap in VLM reasoning, effectively handling tasks requiring visual commonsense and complex reasoning beyond just metaphor understanding.

---

> ### Author Response · Authors · 2025-11-22
> **Response to Reviewer 1K1R (4/n)**
>
> **References:**
>
> [1] Chen et al. MLLM-as-a-Judge: Assessing Multimodal LLM-as-a-Judge with Vision-Language Benchmark. ICML (Oral), 2024.
>
> [2] Gu et al. A Survey on LLM-as-a-Judge. Arxiv: 2411.15594, 2024.
>
> [3] Zhang et al. Can MLLMs Understand the Deep Implication Behind Chinese Images? ACL, 2025.

---

> ### Author Response · Authors · 2025-11-26
> **Kindly Reminder to Reviewer 1K1R**
>
> Dear Reviewer 1K1R:
>
> Sorry to bother you. Thank you once again for dedicating your valuable time to reviewing our paper. We have carefully addressed all your concerns in detail and hope that you find the response satisfactory. We sincerely hope you can participate in the Reviewer-Author discussion, as it is crucial for us to fully address your concerns and improve our paper during this period. As the deadline approaches, we kindly request your feedback on our rebuttal. We are eager to engage in further discussion and address any additional concerns you may have. We sincerely appreciate your constructive suggestions and believe that the additional experiments, analysis, and explanations significantly improve the quality of our submission. We truly hope you might consider raising your score. Thank you!
>
> Best Regards,
> Authors

---

### Author Response · Authors · 2025-12-03
**Summary of Rebuttal Updates & Reviewer Discussion Status**

Dear Area Chair,

We welcome you to this submission. We understand the recent administrative challenges and appreciate the time you are taking to review our work.
We thank the reviewers for their constructive feedback, which has significantly strengthened our work. Following the rebuttal and active discussion period, **Reviewer KHoM and Reviewer eB6x have engaged positively**, and we have successfully addressed the core concerns regarding generalization, diminishing returns on frontier models, and benchmark scale. Notably, **Reviewer KHoM raised their score** after reviewing our new experiments. We have also provided comprehensive responses and new experimental evidence to address the questions raised by Reviewers 1K1R and jGQm.

We summarize the key highlights and resolved issues below:

### 1. Robust Generalization Beyond Image Implication
A primary concern (Reviewer KHoM, jGQm) was whether the LAD framework was a niche solution or a generalizable reasoning architecture.
* **Action:** In the discussion phase, we deployed the LAD framework (without modification) on **general multi-modal benchmarks**: MMMU (Expert AGI), SeedBench, and MMStar.
* **Result:** LAD consistently improved performance across all general tasks. Most notably, **LAD enabled the lightweight GPT-4o-mini to surpass the much larger GPT-4o on SeedBench (77.2 vs 76.7)**. This proves LAD is a fundamental cognitive enhancer for complex VQA, not just a metaphor specialist.

### 2. Eliminating "Diminishing Returns" on Frontier Models (Gemini 3.0)
Reviewer KHoM asked if the "contextual gap" LAD addresses would simply disappear with newer, larger models.
* **Action:** We benchmarked LAD against the newly released **Gemini-3.0-Pro** (Nov 2025).
* **Result:**
    1.  **Cost-Efficiency:** `GPT-4o-mini + LAD` outperforms the base `Gemini-3.0-Pro` on the English Open-Style Question task (4.02 vs 3.82).
    2.  **Additive Gains:** We applied LAD *on top of* Gemini-3.0-Pro. This yielded further significant gains (English MCQ: 76% $\to$ **82%**; OSQ: 3.82 $\to$ **4.30**).
* **Conclusion:** This conclusively demonstrates that scaling alone does not solve the "contextual missing" problem. LAD provides structured reasoning gains that remain complementary to even the strongest frontier models.

### 3. Scalability and Benchmark Rigor
Reviewers (KHoM, eB6x) initially queried the small sample size (100 images).
* **Action:** We have conducted generalization experiments on the **full II-Bench (1,399 images)** and **CII-Bench (800 images)** in the Appendix.
* **Result:** LAD achieved **SOTA performance** on these large-scale datasets (2,000+ images), with `GPT-4o-mini + LAD` outperforming `GPT-4o` and `Claude-3.5-Sonnet` on open-style generation tasks. This confirms our results are statistically robust.

### 4. Reviewer Consensus
* **Reviewer KHoM  (Score raised positively):** Initially skeptical about novelty and generalization. After seeing the MMMU/SeedBench results and the Gemini 3.0 experiments, they stated: *"The additional experiments on tasks other than OSQ have strengthened my view... I will raise my rating."* The reviewer is now convinced of the method's effectiveness.
* **Reviewer 1K1R (Score 4) & jGQm (Score 4):** While we have not received further feedback from these reviewers during the discussion period, the exact concerns they raised (Generalization and Frontier Baselines) were the primary focus of our new experiments. Given that these same results successfully convinced Reviewer KHoM, we are confident that we have fully addressed the underlying issues behind 1K1R and jGQm's ratings.
* **Reviewer eB6x (Score 6):** Remains positive, praising the framework as "creative and well-structured" and effective at handling culturally dependent metaphors.

### Conclusion
We have demonstrated that **Let Androids Dream (LAD)** is a model-agnostic, scalable, and highly generalizable framework. It allows lightweight models to outperform frontier models on complex reasoning tasks and provides additive value to even the newest SOTA models (Gemini 3.0). We believe we have fully resolved the reviewers' reservations.

We sincerely hope that our meticulous response to reviewer comments will facilitate a fair and comprehensive assessment of our work.

---

### Meta-Review · Area_Chair_9G4z · 2026-01-02

**Summary:**

AI still struggles to understand images metaphorically, as existing models have difficulty grasping cultural, emotional, and contextual nuances. While MLLM excels in VQA, it is limited in its ability to understand image implications due to contextual gaps that prevent it from identifying visual relationships and abstract meanings. This paper therefore proposes the Let Androids Dream (LAD) framework to improve image implication understanding and reasoning. LAD addresses missing contextual pieces through three stages: (1) Perception: Converting visual information into multi-layered textual representations; (2) Search: Iteratively searching and integrating cross-domain knowledge to resolve ambiguity; and (3) Reasoning: Generating contextually coherent implications through explicit reasoning.

Initial concerns centered on perceived weaknesses in novelty and academic contribution, verification routing, robustness, evaluation reliability, cost/latency, and generalizability/bias. The authors' rebuttal presented additional experiments and explanations that partially addressed several practical concerns. However, the theoretical foundations cited "dual-process theory (System 1/2) and active information seeking theory" fail to clearly define what the novel algorithm specifies, guarantees, or predicts in the context of machine learning. This raises the risk that the algorithm appears to be merely a rephrasing of existing RAG/CoT approaches. Therefore, we recommend rejecting it at this stage.

**Reviewer Concerns:**

Reviewers raised the following concerns:

Specifically, they noted weak novelty and academic contribution, insufficient verification of routing (model knowledge versus web knowledge), inadequate assessment of pipeline vulnerability and robustness, insufficient reliability and comprehensiveness of the evaluation, unclear cost and latency, and impractical implementation guidelines. They also expressed concerns about generalizability and bias.
Through their rebuttal, the authors provided additional experiments and explanations, and it is believed that many of the above concerns have been addressed.

However, regarding the theoretical foundations that many reviewers raised as concerns, the authors responded that their implementation is based on "dual-process theory (System 1/2) and active information-seeking theory." As a machine learning paper, however, it is unclear what constitutes the "novel algorithm," what it guarantees, and what it predicts. Consequently, the paper is highly likely to be perceived as merely "rephrasing existing RAG/CoT in cognitive terms."

**Reviewer Scores:**

Reviewer 1K1R has a score of 4 and has not participated in the discussion.

Reviewer KHoM commented on raising the score, but the original score was a 2, so there is still distance to go to reach a positive score of 5.

Reviewer eB6x has a score of 6 and said they would keep it.

Reviewer jGQm has a score of 4 and has not participated in the discussion.

Both reviewers, 1K1R and jGQm, scored a 4 and did not participate in the discussion; however, both expressed concerns about the novelty of the proposed method. As stated in the "Reviewer Concerns" section, the authors' response regarding novelty is "reasonably plausible to a certain extent," but it remains insufficient to "decisively" dispel the concerns. Therefore, it is unclear whether these reviewers increased their scores after the discussion.

---

### Decision · Program_Chairs · 2026-01-26

Reject